# Reaction-passivation mechanism driven materials separation for recycling of spent lithium-ion batteries

Zihe Chen[1], Ruikang Feng[1], Wenyu Wang[1], Shuibin Tu[1], Yang Hu[1], Xiancheng Wang[1], Renming Zhan[1], Jiao Wang[1], Jianzhi Zhao[2], Shuyuan Liu[2], Lin Fu[1] & Yongming Sun [1] ✉

Development of effective recycling strategies for cathode materials in spent lithium-ion batteries are highly desirable but remain significant challenges, among which facile separation of Al foil and active material layer of cathode makes up the first important step. Here, we propose a reaction-passivation driven mechanism for facile separation of Al foil and active material layer. Experimentally, >99.9% separation efficiency for Al foil and $LiNi_{0.55}Co_{0.15}Mn_{0.3}O_2$ layer is realized for a 102 Ah spent cell within 5 mins, and ultrathin, dense aluminum-phytic acid complex layer is in-situ formed on Al foil immediately after its contact with phytic acid, which suppresses continuous Al corrosion. Besides, the dissolution of transitional metal from $LiNi_{0.55}Co_{0.15}Mn_{0.3}O_2$ is negligible and good structural integrity of $LiNi_{0.55}Co_{0.15}Mn_{0.3}O_2$ is well-maintained during the processing. This work demonstrates a feasible approach for Al foil-active material layer separation of cathode and can promote the green and energy-saving battery recycling towards practical applications.

Lithium-ion batteries (LIBs) are the dominating power sources for electric vehicles and are penetrating into the large-scale energy storage systems[1,2]. After 5–10 years' service, the accumulated end-of-life LIBs are estimated up to 464,000 tons in 2025 over the world[3–6], which makes their recycling an urgent task with respect to resource and environmental consideration[7–9]. Cathode active materials of spent LIBs contain valuable metals (e.g., lithium, nickel and cobalt, etc)[10] and various approaches have been developed for their recycling, such as the recovery of valuable metals through classical pyrometallurgy[11], hydrometallurgy[12] and bio-metallurgy[13] techniques, as well as the emerging direct regeneration[14,15]. Among all the operations, separation of active material layer from Al foil of cathode becomes one of the most important procedures. The subsequent recycling of active material and Al foil can be readily promoted after their efficient separation.

Currently, several strategies for Al foil-active material separation include high-temperature treatment, dissolution of binder in cathodes using organic solvents and dissolution of Al foil with mineral acids (Fig. S1). High-temperature treatment in air can cause the decomposition of binder and the consequent bond failure between the active material layer and Al foil. Its easy operation makes the high-temperature treatment current mainly adopted separation approach, which, however, faces with high energy consumption and inevitably hazardous emission (e.g., hydrogen fluoride, phosphorus oxide and other toxic compounds), as well as introduction of abundant Al residuals in the separated active material. Dissolution of binder in cathode using certain solvent could be an alternative approach for Al foil-active material separation. Typically, one can employ N-methyl-2-pyrrolidone (NMP), N, N-dimethylformamide (DMF), N, N-dimethylacetamide (DMAC) and dimethyl sulfoxide (DMSO) to dissolve the mostly widely used polyvinylidene fluoride (PVDF) binder in cathode. However, owing to the limited solubility of these solvents for PVDF (5–10 wt%), large number of solvents should

[1]Wuhan National Laboratory for Optoelectronics, Huazhong University of Science and Technology, Wuhan 430074, China. [2]Mirattery Co., Ltd., Wuhan, China. ✉e-mail: yongmingsun@hust.edu.cn

be involved and mechanical assistance often be required. Al foil-active material separation could also be realized through chemical corrosion of Al foil[16]. The mineral acid could chemically dissolve Al foil, leading to rapid separation between active material layer and Al foil[17]. Nevertheless, continuous reaction of Al foil and mineral acid would produce numerous hydrogen gas and dissolution of Al into the reaction solution, as well as the dissolution of active material. All the above-mentioned strategies would unavoidably introduce Al residuals (1–3 wt%) into active material, which cause trouble for the subsequent recycling procedure[18]. Therefore, it is highly desirable but challenging to develop facile approaches based new mechanism for efficient and environment-friendly separation of active material and Al foil with low energy consumption.

Herein, we proposed a reaction-passivation driven mechanism for Al foil-active material layer separation. Experimentally, aqueous phytic acid (PA), with six phosphate carboxyl and twelve hydroxyl groups, was employed as the reagent for cathode separation of LIBs (Fig. S2). As showed in Fig. 1a, the strong acidity of PA can induce its fast reaction with surficial $Al_2O_3$ and metallic Al on Al foil to produce $Al^{3+}$ ions and bubbles (eq. 1), which lead to the loss of contact between cathode active material layer and Al foil[19]. Active material layer was adhered to Al foil in cathode through weak van der Waals interaction between PVDF and Al foil[20]. PA reacted with surficial $Al_2O_3$ and metallic Al on Al foil to produce a dense Al-PA layer with strong covalent bond interaction with Al foil[21], which, together with bubbling, could damage the interaction between Al foil and PVDF in active material layer. The cathode material layer was then facilely separated and collected by physical sedimentation due to their large size (several centimeters), which were beneficial for further operations. The PA molecule would immediately chelate with $Al^{3+}$ to form aluminum-phytic acid complex (Al-PA) (eq. 2) and terminate the further corrosion reaction between

PA and surficial Al. In principle, an $Al^{3+}$ ion can coordinate with 1–3 PA molecules, and a PA molecule can coordinate with certain amounts of $Al^{3+}$ ions via strong chelating ability from its rich phosphate/carboxyl groups (Fig. 1b)[22,23]. Due to above various connectivity patterns between PA and $Al^{3+}$ ions, an intricated [Al-PA] network would be produced in-situ once tiny amount of surficial Al be dissolved. Thus, PA solution could minimize corrosion via in-situ passivation mechanism and shows great potential for efficient separation of active material layer and Al foil with low cost and energy consumption for highly efficient recycling of battery cathode materials (Fig. 1c)[24].

## Results

### Operations of Al foil-active material separation

The reaction-passivation driven Al foil-active material layer separation was monitored by in-situ optical microscope investigation (Fig. 2a). $LiNi_{0.55}Co_{0.15}Mn_{0.30}O_2$ (Ni55) layer was completely separated from the Al foil in only 5 mins (Figs. S3–5 and Supplementary Movie 1). To reveal the practical feasibility, Al foil-active material layer separation experiment was further successfully demonstrated for Ni55 cathode with a total mass of ~705 g from a retired 102 Ah spent cell (Figs. S6–10 and Table S1). Due to the quick penetration of PA solution into the active material layer (Fig. S11), the Ni55 layer was completely peeled off from Al foil (2 × 11.5 m) in 5 mins (Fig. 2b). Interestingly, the obtained Al foil demonstrated a clean surface without any residual active material and observed damage (Figs. 2c and S12). Also, the thickness of the separated Al foil (denoted as S-Al foil) was the same as that of the initial one (13 μm), suggesting that the good stability of Al foil in PA solution (Fig. S13). More elaborate measurement was further subject to inductively coupled plasma mass spectrometry (ICP-MS) measurement and the result revealed an ultra-low Al content of 1.85 wt% in the used PA solution. The above results verified that Al foil could be passivated

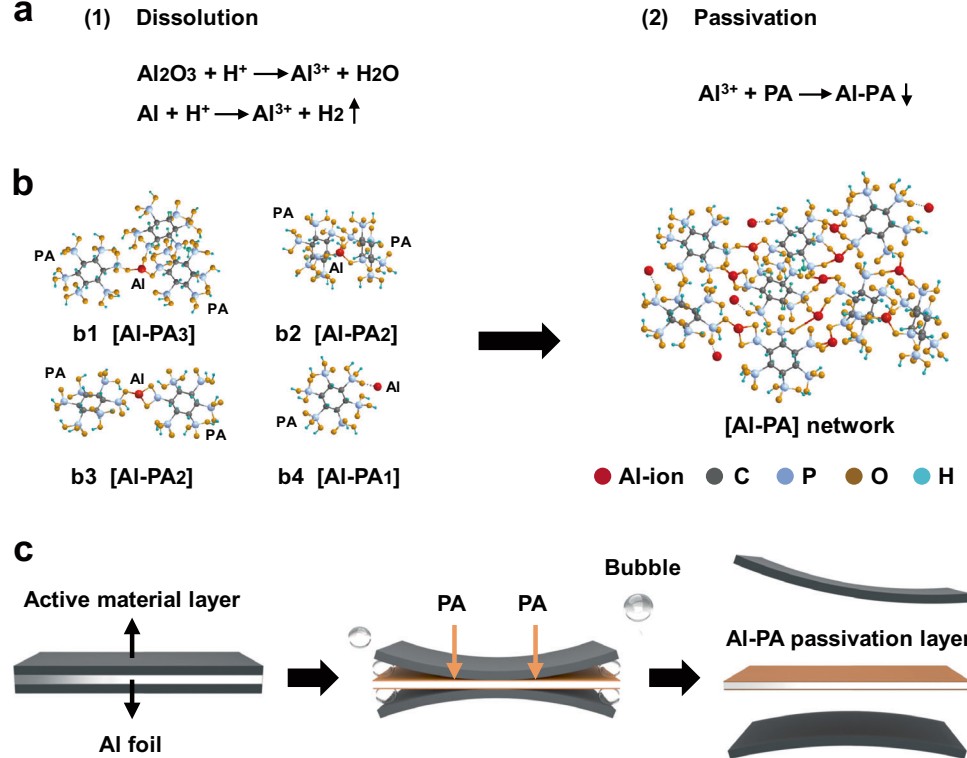

**Fig. 1 | Schematic of reaction-passivation driven separation of Al foil and active material layer. a** The reaction-passivation mechanism of Al foil with PA. **b** The plausible connectivity between PA and Al ions. The b1 and b2 show the chelation between one Al ion with three phosphate groups from three and two PA molecules, respectively. The b3 illustrates the chelation between one Al ion and two phosphate groups from two PA molecules. The b4 displays the chelation between one Al ion and one phosphate group from one PA molecule. **c** Schematic of the separation process between active material and Al foil in PA solution.

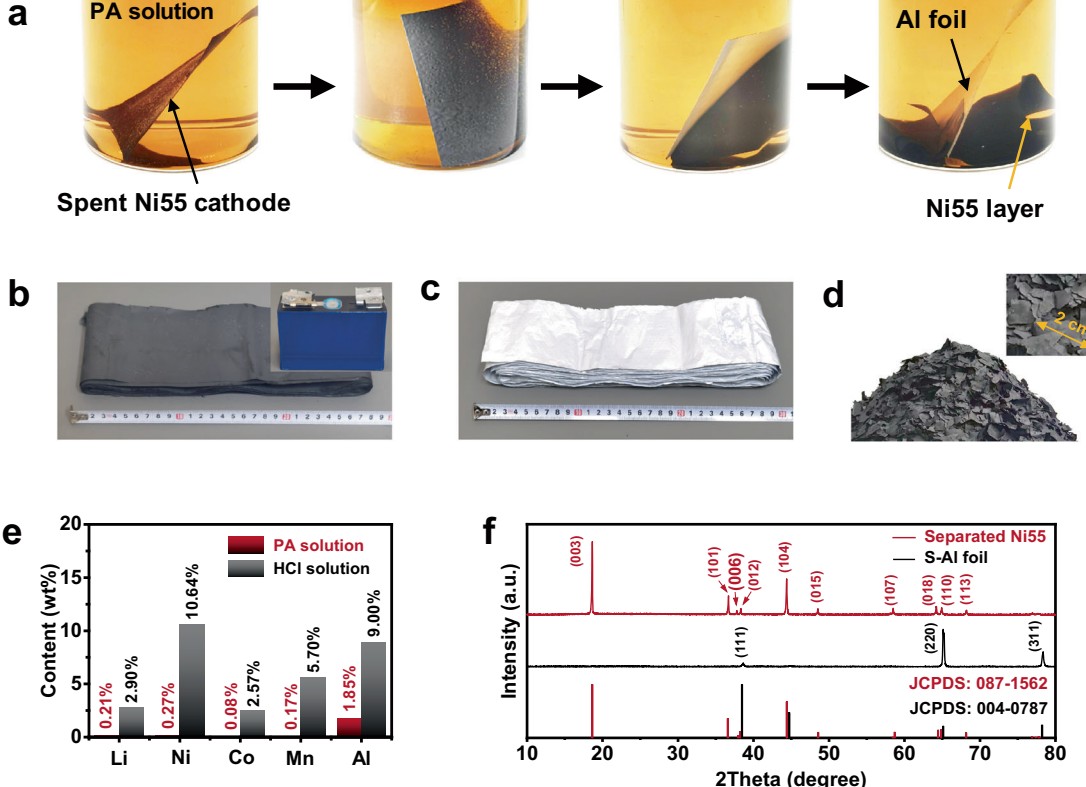

**Fig. 2 | Separation of Al foil and active material layer of Ni55 cathode. a** Digital images of Al foil-Ni55 layer separation process in PA solution. **b** 11.5 m-length cathode of a 102 Ah spent cell. The inset showed the digital image of a 102 Ah spent cell for Al foil-Ni55 layer separation. **c** 11.5 m-length Al foil and **d** Ni55 layer after their separation. **e** Al, Li, Ni, Co and Mn contents in PA solution after Al foil-Ni55 layer separating operation. As a control, the separation experiment of Ni55 cathode was also conducted using HCl solution, and the Li, Ni, Co and Mn contents in HCl solution was measured. **f** XRD patterns of the separated Al foil and Ni55 layer. The JCPDS of 087-1562 and 004-0787 were referred to confirm the layered lithium nickel cobalt manganese oxide and metallic Al, respectively.

instead of continuous corrosion in PA solution after the initial reaction for Al foil-active material layer separation. The contents of Li, Ni, Co, Mn and Al in the S-Al foil was further measured by ICP-MS and the residual Ni55 on the S-Al foil was calculated as only 0.0035 wt%, indicating an ultrahigh Al foil-active material layer separation efficiency of above 99.9 % (Table S2). The separated Ni55 layer demonstrated irregular pieces with size of several centimetres after its separation from the Al foil in PA solution, which could be easily collected by physical sedimentation, and Ni55 pieces with a total mass of ~642 g was finally obtained (Fig. 2d). The losses of Li, Ni, Co, and Mn were only 0.21, 0.27, 0.08, and 0.17 wt% based on the total mass of the Ni55 cathode. The Al residues in the separated Ni55 were as low as 0.026 wt%, again supporting the passivation of Al foil during the processing (Fig. 2e and Table S1). As a contrast, without the protection from in-situ formed Al-PA layer, Al foil was be fully leached in HCl solution, while Ni55 was damaged obviously with a high total Li, Ni, Co and Mn loss of 21.8 wt% (Figs. 2e and S14–16). Thus, the as-explored reaction-passivation driven Al foil-active material layer separation can minimize the corrosion of Al foil and damage of Ni55 during the separation.

X-ray diffraction (XRD) measurement was further conducted to investigate the separated Ni55 material. As shown in Figs. 2f and S17, high ratio of I(003)/I(104) and clear split of the (018)/(110) and (006)/(012) peak pairs were shown, demonstrating the well-defined hexagonal-NaFeO₂ structure of the separated Ni55, which was close to that before PA treatment[25,26]. Pure phase of metallic Al was also verified for the S-Al foil by XRD. Therefore, good integrity was realized for Al foil and active material layer via the as-employed reaction-passivation mechanism, which showed great advantages for further processing of material regeneration.

## Mechanism of Al foil-active material separation

During the Al foil-Ni55 layer separation, PA reacted with the surficial Al₂O₃ and metallic Al to produce an ultrathin, dense Al-PA layer on Al foil, which could terminate their further reaction (Fig. 3a). The composition and chemical state of the Al-PA layer on S-Al foil was then studied by Fourier transform infrared spectroscopy (FTIR) and X-ray photoelectron spectroscopy (XPS). In comparison to fresh Al (F-Al) foil, new stretching vibration peaks at 1162, 1438 and 1641 cm⁻¹ were shown in the FT-IR spectrum of S-Al foil (Fig. 3b), corresponding to $PO_4^{3-}$, phytate and $HPO_4^{2-}$ (consisting of P-O bonding), respectively[27,28]. The bonding between O, P and Al was then verified by the peaks at 75.6 eV in high-resolution Al 2$p$ spectrum, 531.6 eV in high-resolution O 2$s$ spectrum, and 135.2 eV in high-resolution P 2$p$ spectrum (Fig. S18), suggesting the formation of Al-PA layer on the surface of S-Al foil[29–31]. The thickness of Al-PA layer on S-Al foil was analysed by XPS depth detection investigation on P-element content changes during argon plasma etching (Fig. 3c). During the initial sputtering of the sample, the intensity for P-O peak in high-resolution P 2$s$ spectrum decreased gradually from 0 to 90 s, remained stable from 90 to 120 s, and disappeared after 120 s. The thickness of Al-PA layer was then estimated as ~20 nm. The formation of dense Al-PA layer was also verified by the results of Auger electron spectroscopy (AES), high resolution transmission electron microscope (HRTEM) and the corresponding energy dispersive spectrometer (TEM-EDS) mapping measurements (Figs. S19, 20).

To show the effect of Al-PA layer on suppressing the corrosion of Al foil, potentiodynamic polarization test was conducted for F-Al and S-Al foil. The S-Al foil showed higher Ecorr and lower Icorr values compared to that of the F-Al foil (−0.88 vs. −0.92 V, −1.95 vs.

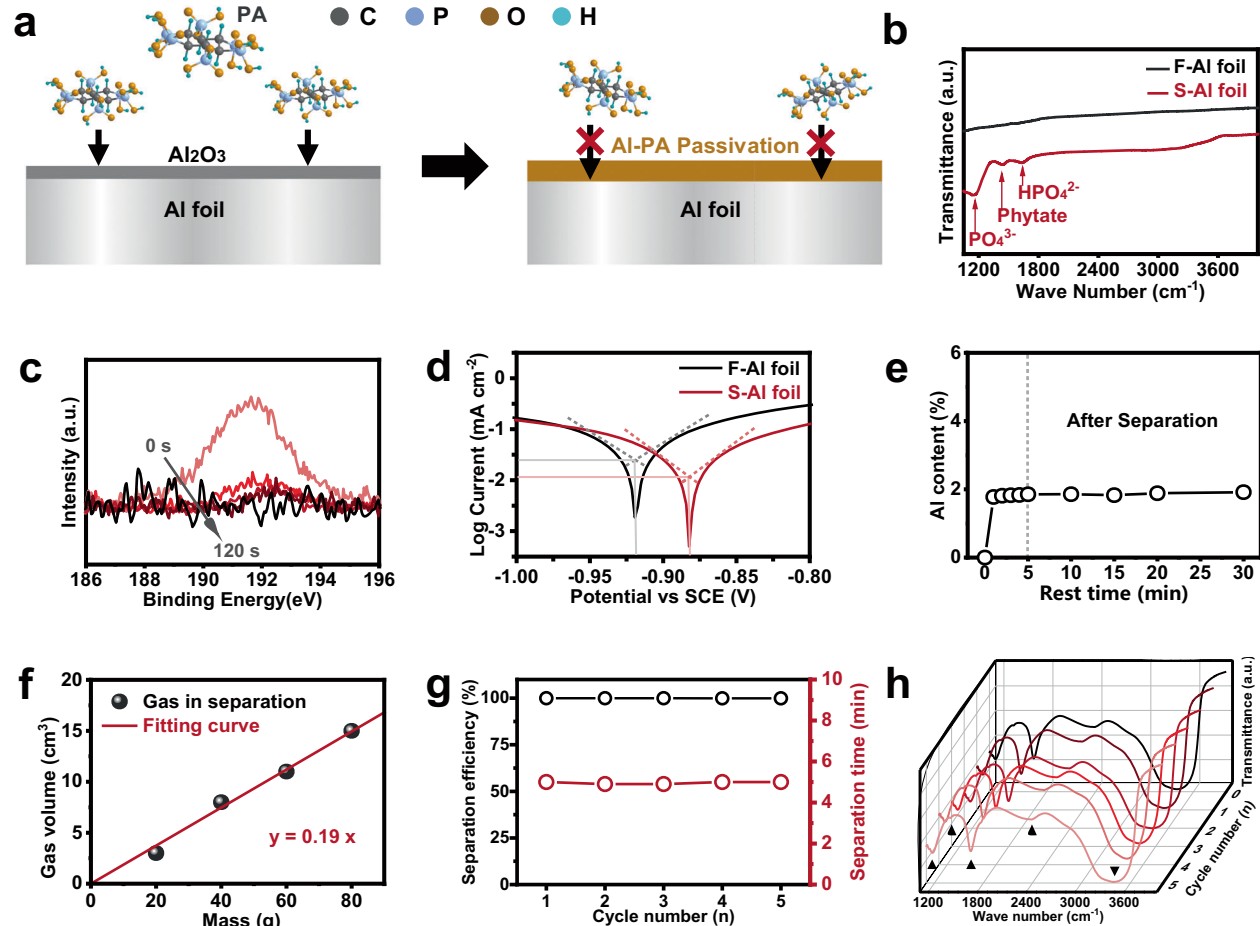

**Fig. 3 | Characterizations of Al foil and PA solutions under different conditions.** **a** Schematic of the reaction-passivation mechanism for Al foil-Ni55 layer separation. During the separation, PA reacted with the surficial $Al_2O_3$ and metallic Al to produce an ultrathin, dense Al-PA layer on Al foil, which could terminate the continuous corrosion of Al foil. **b** The FT-IR spectra of F-Al and S-Al foils. **c** High-resolution P 2 *s* XPS spectra of S-Al foil upon the Ar$^+$ sputtering. **d** Tafel curves of F-Al and S-Al foils. **e** The Al contents in PA solution caused by the dissolution of Ni55 during the Al foil-Ni55 layer separation process. **f** Amount of gas production for Al foil-Ni55 layer separation with different masses from 20 to 80 g. **g** The separation efficiency vs. cycle number plot for Al foil-Ni55 layer separation with repeated use of the same PA solution. **h** FT-IR spectra of above repeated used PA.

−2.05 mA m$^{-2}$, Fig. 3d), suggesting that the S-Al foil was passivated by the Al-PA layer[27]. To further confirm the effect of the Al-PA layer, electrochemical impedance spectroscopy (EIS) was conducted and the values of derived charge transfer resistance (Rct) from the solution to the tested samples were compared. As shown in Fig. S7 and Table S3, S-Al foil displayed doubled value of Rct in comparison to that of F-Al foil, again verifying the well passivated surface structure of S-Al foil by Al-PA layer[32]. Thus, an ultrathin, dense Al-PA layer was formed during the Al foil-Ni55 layer separation, which effectively protected Al foil from continuous corrosion[33].

Another important parameter for the verification of reaction-passivation mechanism is the Al content change in PA solution on time, on weight of cathode and recyclability of PA solution. We monitored the Al content change in PA solutions during the Al foil-Ni55 layer separation process using ICP-MS (Fig. 3e). The Al content in PA solution quickly increased from 0 to 1.8 wt% (based on the mass of spent Ni55 cathode) in 1 min, and then remained constant. Importantly, such a low Al content kept unchanged even after the complete Al foil-Ni55 layer separation (5 mins). The measured value of Al content was 1.9 wt% after further 25 mins' resting of Al foil in PA solution. This result supported the quick formation of dense Al-PA layer on Al foil surface, which inhibited the continuous Al dissolution. The formation of stable, ultrathin Al-PA layer was further evidenced by the results of XPS

investigation on Al foils with different PA treatment times (Fig. S21). To verify the capability of producing stable Al-PA layer, the as-formed Al-PA layer was polished and then treated with fresh PA solution under the same condition as the initial treatment. This PA solution showed similar Al$^{3+}$ concentration to that for the initial Al foil-active materials layer separation (0.472 wt% and 0.478 wt%, respectively). Thus, the corrosion of Al foil was significantly suppressed once the formation of Al-PA layer on the surface of Al foil. We further investigated the gas production for Al foil-Ni55 layer separation with different sample masses from 20 to 80 g (Fig. 3f). A linear relation between mass and gas production was observed, and the constant gas production rate was calculated as ~0.19 ml g$^{-1}$. The ultrathin Al-PA passivation layer and the low residual Al content in used PA solution supported the ultralow loss of PA solution, indicating its recyclability. Same PA solution was repeatedly used for Al foil-Ni55 layer separation experiments (five times). The time for separation was constant (~5 mins) and the separation efficiency remained above 99.9% for all the experiments (Fig. 3g and Table S2). After repeated use of PA solution for five times, it displayed the same characteristic vibration bands as the pristine one (Fig. 3h). The above result indicated the constant reaction dose of PA for Al foil and the robust reaction-passivation mechanism driven Al foil-Ni55 layer separation, making it promise for scalability and sustainability. Such active material-Al foil separation approach was also

employed for the separation of Al foil and active material layer in other cathodes including $LiCoO_2$ and $LiFePO_4$ (Figs. S22–27), and facile separation was realized for both the cathodes with high separation efficiency and low PA consumption, which further supported the importance and impact of our PA-direct approach.

## Regeneration of active material

Figure S28 showed the dissolved of Li, Ni, Co and Mn contents from the degraded Ni55 after its separation with Al foil in PA solution, and the results indicated that the dissolution of these elements was negligible (e.g., 0.27% for Ni, 0.08% for Co, and 0.17% for Mn). Separated Ni55 with well-maintained structure and composition provided the basis for facile regeneration via direct annealing of its hybrid with Li salts (Fig. 2f and Table S1). Ni55 cathode from spent cell before Al foil-active material layer separation operation was denoted as degraded Ni55 for simplification. As shown in Figs. S29, 30, scanning electron microscope (SEM) result revealed that the cracks in the degraded Ni55 particles cured after their recovery. Besides, HRTEM and XRD results verified the pure phase of layered α-$NaFeO_2$ structure with R-3m space group for the regenerated Ni55 (Figs. 4a, b and S31, 32)[34]. Its (108) and (110) peaks shifted towards each other and (101) peak moved to a lower degree, suggesting the transformation from cation disorder to order arrangement and thus successful materials recovery (Fig. S33)[35]. As shown in Fig. 4c, the regenerated Ni55 displayed much higher discharge capacities than that before regeneration (166 vs. 152 mAh g$^{-1}$ for Ni55 before and after the regeneration). Moreover, the

regenerated Ni55 delivered high discharge capacities of 161, 157, 146 and 132 mAh g$^{-1}$ at 0.3, 0.5, 1.0 and 2.0 $C$, respectively, far outperforming the counterpart before regeneration (e.g., 91 mAh g$^{-1}$ at 2.0 $C$, Fig. 4d). Also, high capacity retention of 94 % was realized after 100 cycles for the regenerated Ni55 at 0.3 $C$ (Fig. 4e). The successful regeneration of the degraded Ni55 was also evidenced by the results of EIS, constant-current charge/discharge and cyclic voltammetry test (Figs. S34–38 and Tables S3, 4)[36].

## Environmental and economic analysis of PA involved regeneration routine

Procedures for different recycling approaches were schematically shown, including PA involved direct recycling (PA-direct, Figs. 5a and S39), general direct recycling (General-direct, Fig. S40), pyrometallurgical recycling (Pyro, Fig. S41) and hydrometallurgical recycling (Hydro, Fig. S42). The EverBatt model developed by the Argonne National Laboratory was used for the life cycle assessment (LCA) and techno-economic analysis (TEA) of the above recycling processes based on the treatment of 10,000 tons of spent Ni55 cells (Table S5)[37]. Without the need of high energy consumption for cathode pretreatment in the industrial recycling approaches (e.g. shredding, milling/thermal treatment, and sieving), the total energy consumption of the PA-direct was 5.84 MJ kg$^{-1}$ cell (4.17 and 1.67 MJ kg$^{-1}$ corresponding to materials use and processing, respectively), which was much lower than the General-direct, Hydro and Pyro approaches (Fig. 5b and Table S6). Meanwhile, the additional GHG emission for burning mixed

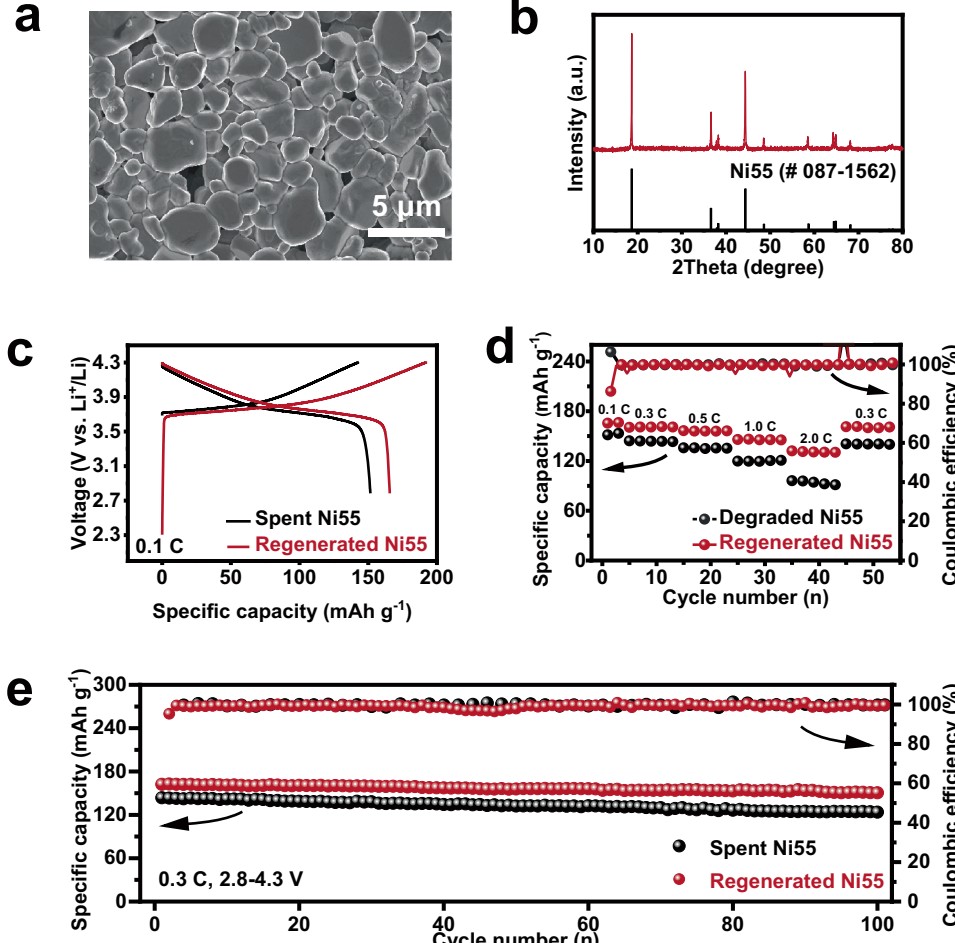

**Fig. 4 | Characterizations of regenerated Ni55. a** SEM image and **b** XRD pattern of the regenerated Ni55. **c** Voltage-capacity profiles for the first cycle at 0.1 $C$, **d** rate

performance at various current densities of 0.1, 0.3, 0.5, 1.0 and 2.0 $C$ and **e** cycling performance at 0.3 $C$ for the degraded and regenerated Ni55.

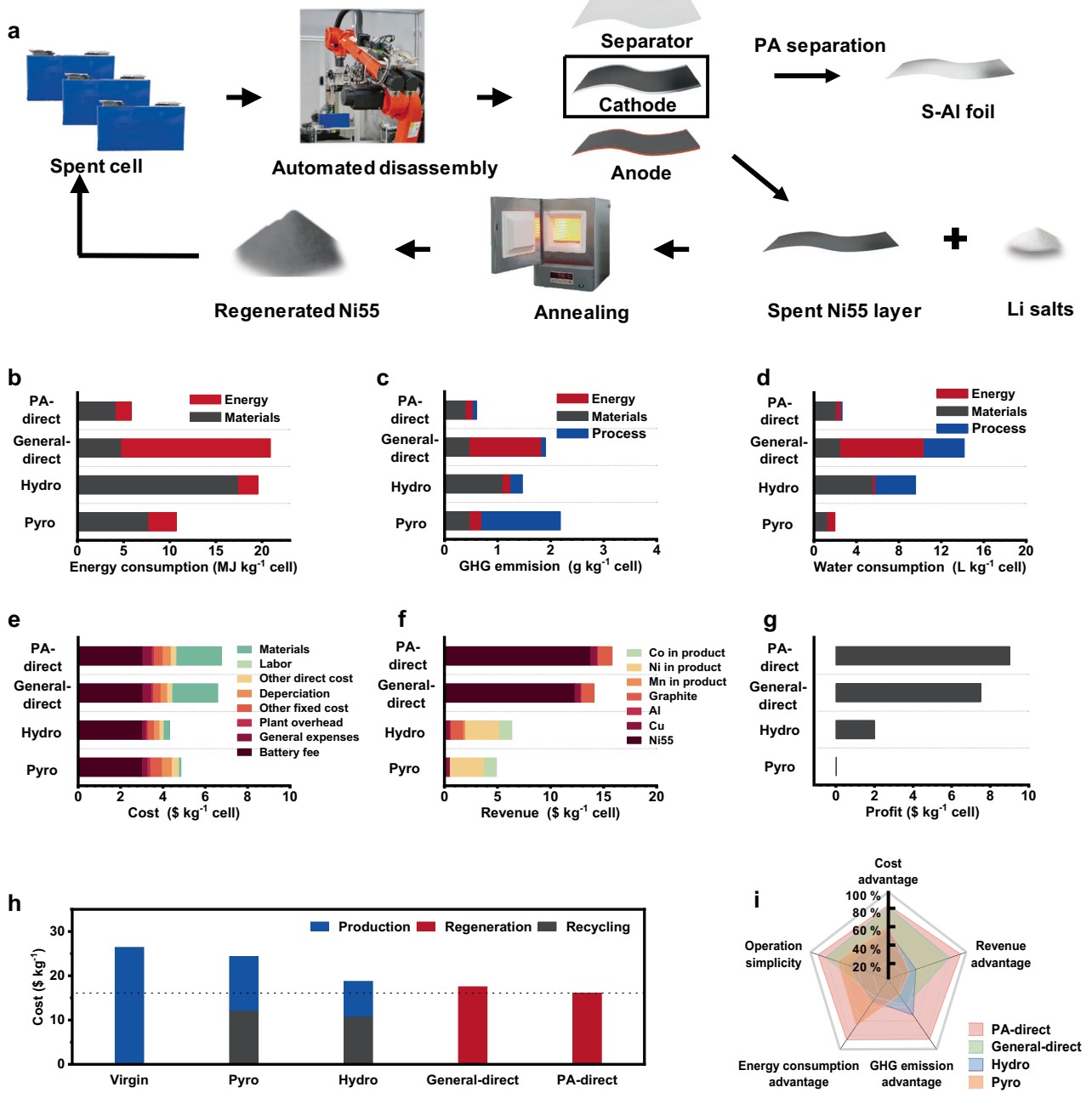

**Fig. 5 | Economic and environmental analysis of PA-direct and other recycling approaches. a** Brief schematic of the PA-direct. **b** Energy consumption, **c** GHG emission, **d** water consumption, **e** cost, **f** revenue and **g** profit for PA-direct, General-direct, Pyro and Hydro. **h** The overall cost of manufacturing 1 kg-Ni55 cathode from raw and recycled materials. **i** Comprehensive comparison of different recycling approaches.

graphite and smelting Al and Cu scarps were avoided in comparison to other recycling approaches due to the advantage of complete separation of cathode from other battery components in our PA-direct (Figs. S39 and 43). Thus, the lowest GHG emission of 0.61 g kg⁻¹ was achieved for PA-direct (Fig. 5c and Table S7). PA solution was recyclable, which enabled low consumption of water. Only 2.65 L kg⁻¹ of water was needed for treating one kilogram of cells, which was comparable to the water consumption of the Pyro process (1.97 L kg⁻¹) and much lower than those of the Hydro (9.58 L kg⁻¹) and General-direct (14.17 L kg⁻¹) processes (Fig. 5d and Table S8).

Figure 5e showed the costs of the above different recycling approaches. The use of Li salt for cathode material repair with the PA-direct (from the degraded Ni55 to regenerated Ni55) led to a slightly high cost of 6.78 $ kg⁻¹ (Tables S9–12). PA-direct possessed the

advantage of high separation efficiency, production of cathode material and direct output of high-performance regenerated cathode material (Fig. S44 and Table S13). Thus, it brought higher revenue and net profit (15.79 and 9.01 $ kg⁻¹, respectively) than other recycling approaches (Fig. 5f, g and Tables S5 and 14). We performed the TEA based on manufacturing 1 kg-Ni55 cathode as a reference for industrial production process. Figure 5h-i and Table S15 showed the cost and profit for manufacturing 1 kg-Ni55 cathode from raw materials (Ni salt, Co salt, Mn salt and Li salt) and recycled materials (degraded Ni55). It is noted that more Li salt was needed for the re-synthesis of active cathode materials using the products in Pro and Hydro as raw materials. The cost for Ni55 cathode manufacture via PA-direct was only 16.07 $ kg⁻¹, much lower than 26.41$ kg⁻¹ for synthesis with raw materials (Virgin) and other processes (17.52 $ kg⁻¹ for General-direct,

24.32 \$ kg$^{-1}$ for Pyro, 18.73 \$ kg$^{-1}$ for Hydro, respectively). Thus, the PA-direct for Ni55 manufacture could achieve a high profit of 16.80 \$ kg$^{-1}$, which was ~2.68 times higher than manufacture from raw materials. As a result, our PA-direct provides a promising route for facile separation of Al foil and active material layer, and active materials regeneration, which can promote energy-saving, environmentally friendly and high-value battery recycling towards practical applications.

## Discussion

In this work, a reaction-passivation driven mechanism was proposed for efficient separation of Al foil and cathode active material layer from spent LIBs. Experimentally, 60 g of Al foil and 636 g of Ni55 from a 102 Ah spent cell were facilely separated using a PA solution in 5 mins with >99.9% separation efficiency. Intact structures of the Al foil remained after its separation and the Ni55 showed low dissolution loss of metal ions during the processing. The reaction between PA and Al led to the formation of 20 nm-thickness, dense Al-PA layer on Al foil during the separation, suppressing the continuous corrosion of Al foil. Through direct annealing of hybrid degraded Ni55 and Li salts, the regenerated Ni55 delivered a reversible capacity of 161 mAh g$^{-1}$ and high capacity retention of 94 % for 100 cycles at 0.3 $C$. Our work provides a promising route for facile separation of metallic current collector and active material layer, which is very different from the traditional approaches, and could promote green and energy-saving battery recycling towards practical applications.

## Methods

### Acquisition of spent cathode

The spent cell was first discharged to 2.0 V and then disassembled to obtain the LiNi$_{0.55}$Co$_{0.15}$Mn$_{0.3}$O$_2$ (Ni55) cathode, which was dried overnight at 60 °C for further processing.

### Separation of Al foil and Ni55 layer using PA solution

The Ni55 cathode was immersed in 30 wt% PA solution (cathode: PA = 0.05, m/m) at 25 °C until Al foil and Ni55 layer were separated. Then they were separately collected, washed with deionized (DI) water and dried overnight at 60 °C to obtained separated Al (S-Al) foil and Ni55.

### Separation of Al foil and Ni55 using HCl solution

The Ni55 cathode was immersed in 30 wt% HCl solution (cathode: HCl = 0.05, m/m) at 25 °C until Al foil was dissolved. The as-separated Ni55 layer was washed with DI water and then dried at 60 °C.

### Concentration investigation of PA solution for Al foil-Ni55 layer separation

10 g of Ni55 cathode was immersed in 200 g of PA solutions at ~25 °C with different concentrations (10, 20, 30, 40 and 50 wt%), respectively. After the separation of Al foil and Ni55 layer, they were washed with DI water and dried overnight at 60 °C, respectively.

### Recyclability investigation of PA solution

10 g of Ni55 cathode was immersed in 200 g of 30 wt% PA solution at ~25 °C. Al foil and Ni55 layer were collected after their complete separation. Then the PA solution was reused for another batch of Ni55 cathode separation. The above operations were repeated for five times.

### Al foil-Ni55 layer separation via ultrasonic treatment

10 g of Ni55 cathode was added into 100 mL beaker with 50 g of solvent (DI water or NMP), followed by ultrasonic treatment for 60 mins with the ultrasonic frequency of 50 Hz and electric power of 50 W. Finally, the as-obtained Al foils were washed with DI water and dried overnight at 60 °C.

### Regeneration of Cathode Materials

The obtained Ni55 pieces were crushed into powders and mixed with LiOH (LiOH: Ni55 = 0.15: 1, mol/mol). Then Ni55/LiOH hybrid was heated at 300 °C for 2 h and 750 °C for 8 h in air to produce regenerated Ni55.

### Fabrication of Ni55 cathode

Regenerated Ni55 was mixed with carbon black (CB) and poly-vinylidene fluoride (PVDF) in anhydrous NMP for slurry fabrication. The slurry was cast onto Al foil and dried at 60 °C for 6 h in a vacuum oven to fabricate Ni55 cathode with a mass loading of 3.5 mg cm$^{-2}$. The weight ratio of regenerated Ni55, CB, and PVDF was 8:1:1 in the as-obtained regenerated Ni55 cathode. The thickness, diameter and average active mass loading of Ni55 cathode were 50–52 μm and 10 cm and 3.5-4.0 mg cm$^{-2}$, respectively.

### Characterizations

Optical microscopy studies were carried out on an Olympus BX53 light microscope with DP27 CMOS detector. Scanning electron microscope (SEM) and energy-dispersive X-ray spectra (EDS) measurements were conducted using a Gemini/SEM 300 field-emission SEM instrument. More detailed characterizations on morphology and structure of materials were conducted by focused ion beam electron microscopy (FIB-SEM, Helios NanoLab G3 CX) and transmission electron microscopy (TEM, Talos F200X). Fourier transform infrared spectroscopy (FTIR) spectra were recorded in the range of 900 - 4000 cm$^{-1}$ on Nicolet iS50R. The composition of Ni55 and PA solvent were analyzed by inductively coupled plasma mass spectrometry (ICP-MS, ELAN DRC-e). X-ray photoelectron spectroscopy (XPS) depth profiles were collected on the VG MultiLab 2000 system (Thermo VG Scientific). X-ray diffraction (XRD) was carried out on a PANalytical B.V. Empyrean X-ray diffractometer using Cu Kα radiation at 40 kV and 30 mA. AES measurement were conducted on JEOL JAMP-9510F Auger electron microscopy and the electron source was Schottky field emission.

### Electrochemical measurements

All the electrochemical measurements were carried out using CR2032 coin-type cells assembled in the argon-filled glove box with the oxygen and water contents both below 0.1 ppm unless otherwise specified. Pure lithium foil was used as the counter electrode and Celgard 2300 membrane was used as the separator. The electrolyte was 1 M LiPF$_6$ in mixed solvents of ethylene carbonate (EC) and dimethyl carbonate (DEC) (1:1 in volume) with 5 wt% fluoroethylene carbonate (FEC). The cathode electrodes were prepared using a traditional electrode fabrication route, including slurry mixing, slurry coating, electrode drying and electrode rolling process. The first-cycle charge/discharge curves were collected in the potential range of 2.8–4.3 V at 0.1 $C$ (1 $C$ = 150 mAh g$^{-1}$). The cycling performance of spent and regenerated NCM cathodes were evaluated under 0.1 $C$ for the initial five cycles and then under 0.3 $C$ for the following cycles using galvanostatic mode. The rate performance of spent and regenerated Ni55 cathodes were evaluated under 0.1 $C$ for the initial three cycles and then under 0.3, 0.5, 1.0 and 2.0 $C$ for the following cycles using galvanostatic mode. Cyclic voltammograms (CVs) and electrochemical impedance spectroscopy (EIS) were conducted on a Biologic VMP3 system. The scan rate for CV measurement was 0.1 mV s$^{-1}$. The frequency for EIS measurement ranged from 100 kHz to 100 mHz with an amplitude of 10 mV. The anti-corrosion performance test was carried out using a three-electrode system in 3.5 wt% NaCl solution at 28 °C on a Biologic VMP3 system. Potentiodynamic polarization was tested by linear sweep voltammetry at a scan rate of 20 mV s$^{-1}$. The whole electrochemical measurements were carried out at 25 ± 2°C.

**Separating efficiency calculations**

$$\eta = 1 - m_1 / m_2 \qquad (1)$$

where the $\eta$, $m_1$ and $m_2$ were separation efficiency of Ni55 cathode, mass of residual Ni55 layer on Al foil and total mass of Ni55 layer before separation, respectively.

**LCA and TEA analysis**

The EverBatt model developed by the Argonne National Laboratory was used for LCA and TEA of PA-direct, General-direct, Pyro and Hydro approaches based on the treatment of 10,000 tons of spent Ni55 cells. The flowchart of PA-direct process in EverBatt was performed through customizing recycling process to reflect the changes in the process design. Recycling approach comparison was conducted based on $LiNi_{0.5}Co_{0.2}Mn_{0.3}O_2$ model. The materials requirements for above recycling processes were obtained from the EverBatt model, and that for PA-direct was calculated based on practical condition in our experiments.

The life-cycle energy consumption, water consumption and GHG emission for PA-direct, General-direct, Pyro and Hydro approaches were composed of material, energy, and process emission (Tables S5–8). To estimate the potential benefits, we assumed that recycled components were valuable and could be compensated for the recycling cost. The detailed process-based cost and revenue models were listed in original data of TEA (Tables S10–15).

## Data availability

The data that support the plots within this paper and other finding of this study are available from the corresponding author upon reasonable request.

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

## Acknowledgements

This work is supported by the National Natural Science Foundation of China (Grant No. 52072137). The authors would like to thank the Analytical and Testing Center of Huazhong University of Science and Technology (HUST) for providing the facilities to conduct the XRD, SEM and TEM characterizations. The authors also thank Yan Zhu in Micro and Nano Fabrication and Measurement Laboratory for the support in experimental verification. The authors also thank State Key Laboratory of Materials Processing and Die & Mould Technology for the support in Auger electron spectroscopy measurement. The spent 102 Ah cell is supported by Mirattery Co., Ltd.

## Author contributions

Z.C.: investigation, data curation, methodology and writing—original draft. R.F.: investigation, data curation and methodology. W.W.: investigation, data curation and methodology. S.T.: data curation and editing. Y.H.: data curation. X.W.: data curation. R.Z.: data curation. J.W.: data curation. J.Z.: data curation and methodology. S.L.: data curation. L.F.: data curation and Y.S.: supervision, conceptualization, methodology, writing—review and editing. All authors discussed and contributed to the results.

## Competing interests

The authors declare no competing interests.
