## [Peer Review File · Nature Communications]

Reaction-passivation mechanism driven materials separation for recycling of spent lithium-ion batteriesREVIEWER COMMENTS

Reviewer #1 (Remarks to the Author):

The work describes a reaction-passivation procedure for the recycling of aluminum foil from battery cathodes.

While the work presents an interesting approach, the work seems to be too specific to be of broad significance or impact to Nature Communications. Aluminum, while a valuable commodity, is not a critical element compared to other battery-material components. Furthermore, there are key unanswered questions in the work, especially regarding the mechanism.

As such, the work is not recommended for publication in Nature Communications.

Some more specific comments are given below:

1. The formation mechanism and the status of the PA-Al layer should be more clear and supported experimentally. Can the authors demonstrate the existence of the PA-Al layer on Al foil and their morphology clearly by TEM-EDS mapping?
2. Separation Time dependent PA-Al thickness changes can be a good evidence for the PA-Al layer formation.
3. Miller indices should be displayed in the XRD patterns since the authors discussed them on p.5
4. P7. Line 196, the sentence "XRD result confirmed the highly ordered layered structure" is vague. Should mention the crystal structure instead of the "ordered layered structure"
5. As a control, the authors can polish out the PA-Al layer after the separation and monitor the Al content change in PA solution again to make sure the role of PA-Al layer.
6. The authors should confirm that PA react with the other active materials and form the passivation layer.

Reviewer #2 (Remarks to the Author):

Recycling LIBs is an important topic and the method shown is a novel approach although there are many similar organic acids which have been used. The idea of selective passivation of metals during dissolution is known in metal processing although it has not been applied to LIB batteries. The approach clearly works although there are two obvious issues which the authors have not addressed.

Firstly phytic acid even on large scale bulk purchase is >\$50/kg and since it is a stoichiometric reagent and used up in the delamination process will make the process economics unviable.

There are faster delamination processes reported which do not have stoichiometric reagents e.g. Green Chem., 2021, 23, 4710 – 4715

The process does still not remove the polymer binder. The major contribution to the LCA is the thermal treatment to remove the binder and the cost of the lixiviant to delaminate the electrode. Since Phytic acid is competing against sulfuric acid at \$0.4 /kg it will never compete.

It is a nice paper but the impact is not high enough for Nature Communications. It really needs LCA/TEA and a comparison with competing techniques to increase the impact.

Reviewer #3 (Remarks to the Author):

The rapid demand for lithium-ion batteries is widely envisaged to lead to a significant amount of battery waste. The environmental concerns along with the uncertainties in the supply chains make the development of efficient recycling technologies urgent and necessary. Due to the complex structure and inherent safety issues, spent LIBs have to be subjected to a variety of pre-treatments prior the recovery of electrode active materials through direct recycling, pyrometallurgy, hydrometallurgy, or a combination of those methods. The efficiency of the pre-treatments and especially the separation of the active material from the current collector is of paramount importance in the effort to decrease the impurities content in the recovered active materials and improve the cost and environmental credentials of the recycling process. The proposed process for materials separation through reaction-passivation of the Al current collector addresses this issue by offering a closed loop recycling process where NMC cathode materials can be easily recovered and regenerated. Phytic acid has been widely used to develop environmentally friendly chemical conversion coatings on variety of metal alloys due to its high coordination sites and ability to chelate with many different metal ions. It was also recently applied as precipitant in hydrometallurgical recycling of LiMn_2O_4 batteries. Nevertheless, the proposed utilization of Phytic acid in the pre-treatment stage is an original contribution leading to some notable advantages. The authors present a substantial amount of experimental and analytical data demonstrating the advantages of their idea. However, the manuscript contains numerous mistakes and inconsistencies as well as deficiencies which require additional contributions. Thus, while I do not question the merits of the work, I am afraid the manuscript does not represent a sufficiently high quality to justify publication in Nature Communications. I would like to ask the Authors to consider the following comments:

(1) In the "Introduction" section the author claim that the loss of contact between the cathode active material and the aluminium foil was due to the hydrogen gas bubbling at the interface. In support of this claim, the authors make a reference to the paper of Cao et al [18] in which graphite anodes were separated from Cu current collectors by electrolysis method, where hydrogen, continuously produced at the negative electrode, weakened the bonding force between the binder and the copper foil. However, while the cited electrolysis process was continuously applied for at least 25 min, according to the authors the passivation of aluminium by Phytic acid happens very quickly – 1 min according to Figure 3e. Logically, such passivation would suppress further formation of hydrogen, thus gradually reducing or removing the impact de-bonding force entirely. The mechanism and duration of the hydrogen generation would depend on the kinetics of passivation layer formation, surface coverage etc. It is apparent that Figure 2, as presented, does not suggest continuous bubbling. Hence, the following aspect of the work is not clarified - is the gas generation at the interface the only de-bonding mechanism or could there be contribution from other de-bonding mechanisms?

(2) Figure S6 presents important data on the dynamic of passivation process depending on PA concentrations. However, it is practically not discussed in the manuscript and not supported by additional characterization – e.g. Potentiodynamic polarization and/or Impedance spectroscopy of S-Al. It is well documented in the literature that at high concentration of PA the large number of phosphoryl groups adsorbed the surface of aluminium can hinder the formation of Al-PA complexes while at low concentration the passivation film could be incomplete.

(4) Figures S3 and S4 do not illustrate separation time as claimed in the text on page 4!

(5) The authors make reference to the figures in the Supplementary information without previous explanation or description of the experiment undertaken. This leaves the reader to

deduce the author's intentions from the capture of the figures . In example – reference to Figure S9 presenting schematic diagrams and digital images of the spent Ni55 cathode with covered tap for PA penetration behavior investigation.

(6) The reference to Figure S12 on page 5 is entirely incorrect. Figure S12 illustrates optical microscopy images for Al foils separated by using ultrasonic treatment for 1 h in (a) DI water and (b) NMP solution. This is an experiment not discussed in the main text and certainly not the PA separation commented in the text.

(7) Wrong figure reference on page 5:

“The Al residues in the separated Ni55 were as low as 0.026 wt%, again supporting the passivation of Al foil during the processing (Figure 2e and Table S2).”

It should be referenced to Table S1!

(8) The lack of peaks indexation in Figures 2f and S16 makes the XRD analysis on page 5 unclear.

(9) The acronyms F-Al and S-Al are introduced in the text on pages 5 and 6 without explanation.

(10) The term “degraded Ni55” is introduced without explanation on page 7 and its relation to Figure S19 is not clearly explained.

(11) The data on the electrochemical performance of regenerated and degraded Ni 55 (Figures S24 to S28) should be commented in the text rather than simply referred to.

(12) The caption of Figure S25 does not explain clearly the presented graphs. I guess these are views of the charge curves zoomed to the lower specific capacity values. What was the goal of the authors presenting those?

(13) Simplified economic analysis of the recycling process (cost and availability) is desirable and can be done in comparison to a conventional separation process (e.g. vs HCl leaching).

(14) It is unclear from the text in the manuscript how referencing to sources [32-34] contribute to the claims made by the authors?

Response to reviewers' comments:

General response to the Reviewers

Dear Reviewers,

On behalf of our co-authors, we would like to thank you for reviewing our manuscript, titled “*Reaction-passivation mechanism driven materials separation for recycling of spent lithium-ion batteries*”.

Recycling of spent lithium-ion batteries is urgent with respect to resource and economy, especially for the cathodes that contain valuable metals (such as lithium, nickel and cobalt, etc). Among all the operations for cathode recycling, facile and efficient separation of active material layer from Al foil is one of the most important procedures, after which the subsequent recycling of Al foil and active material can be readily facilitated. However, the reported strategies for Al foil-active material layer separation often faces drawbacks of high energy consumption, inevitably hazardous/greenhouse gas emission, use of abundant chemicals and complex operations. Therefore, it is highly desirable but challenging to develop facile approaches for efficient, environment-friendly separation of active material layer for cathode regeneration. This work explored a reaction-passivation mechanism driven active material layer separation with high efficiency, low energy consumption and low greenhouse gas emission. The de-bonding between Al foil and binder in phytic acid (PA) solution enabled their separation with negligible damage of cathode active materials, which brought great bonus for facile regeneration of the degraded cathode active material. Experimentally, kg-level $\text{LiNi}_{0.55}\text{Co}_{0.15}\text{Mn}_{0.3}\text{O}_2$ (Ni55) cathode was separated with high efficiency (> 99.9%) in 5 mins, and well-maintained structure and composition of the separated Ni55 enabled its facile regeneration towards high-performance cathode active material.

Below is a brief summary of the innovations and significances of our reaction-passivation mechanism driven materials separation for recycling of spent lithium-ion batteries.

- *Innovation #1: A reaction-passivation driven strategy is first proposed for easy separation of Al foil and active material layer using PA solution with low energy consumption and greenhouse gas emission. Ultrathin, dense aluminum-phytic acid complex (Al-PA) layer was in-situ formed on Al foil immediately after its wetting with PA, which terminated their further reaction. De-bonding between active material and Al foil enabled contact failure between active material layer and Al foil for separation.*
- *Innovation #2: Using the reaction-passivation driven strategy, separation of Al foil and active material layer in cathode from a practical cell from electric vehicle was finished in 5 mins at room temperature with separation efficiency >99.9%, demonstrating a recorded-high separation efficiency for kg-level Al foil-active material layer separation.*

- *Innovation #3: The separated cathode active material (e.g., Ni55) with well-maintained structure and low Al impurity provided the great bonus for facile regeneration of degraded active material via direct annealing with the addition of Li salts. The regenerated Ni55 demonstrated comparable electrochemical performance to state-of-the-art fresh Ni55.*

In the past 3 months, we have carefully considered all of your comments, carried out a large amount of experimental work, and significantly revised the manuscript, including the following.

- *To demonstrate the significance and impact of reaction-passivation mechanism driven materials separation for recycling of spent lithium-ion batteries, we conducted the corresponding life cycle assessment (LCA) and techno-economic analysis (TEA) (Reviewer #2 and Reviewer #3). Following Reviewer #2 and Reviewer #3's suggestion, we conducted the LCA and TEA for our PA involved cathode direct recycling approach, as well as its comparison with traditional recycling approaches, to demonstrate the broad impact of this work.*
- *We performed Auger electron spectroscopy (AES), transmission electron microscope (TEM) and energy dispersive spectrometer (TEM-EDS) mapping measurements to verify the existence of dense Al-PA layer on Al (Reviewer #1).*
- *We conducted X-ray photoelectron spectroscopy (XPS) measurement for S-Al foils with different separation times to support the formation of stable and robust Al-PA layer (Reviewer #1).*
- *To show the generality of reaction-passivation mechanism driven Al foil-active material layer separation to different spent lithium-ion batteries, we performed the separation experiments and analysis on other types of cathode active materials (Reviewer #1). To address the reviewer's concern, we conducted the Al foil-active material layer separation experiments for spent LiCoO₂ (LCO) and LiFePO₄ (LFP) cathodes using the as-proposed reaction-passivation mechanism driven separation.*
- *We carried out the analysis on the passivation of Al foils after treatment in PA solutions with different concentrations (Reviewer #3). Following the reviewer's suggestion, we measured potentiodynamic polarization and impedance spectroscopy for separated Al foils in PA solutions with different concentrations to analyze the passivation effect of Al foil in PA solution.*
- *We added more detailed experimental methods and data, as well as references to the manuscript and supplementary information (Reviewer #1, Reviewer #2 and Reviewer #3).*

We appreciate all the constructive comments from the three reviewers. We have greatly improved the manuscript by including these extensive new data in the revised manuscript (including the supporting information).

Our work reports a novel reaction-passivation mechanism driven separation of Al foil and active material layer of cathodes for battery recycling. Technically, facile separation of kg-level Ni55 layer and Al foil in cathode for a practical cell from electric vehicles was demonstrated, featuring fast separation rate, ultrahigh separation efficiency, low energy consumption, greenhouse gas emission, intact Al foil and well-maintained Ni55 with low Al impurity after their separation. Our finding may pioneer a new research direction of separation strategy exploration for degraded electrode of end-of-life lithium ion batteries and offer significant insight into efficient, green and energy-saving battery recycling.

As a result, we believe that the significance of this work should be of broad interest and is appropriate for *Nature Communications*. On behalf of my co-authors, we thank you for your thoughtful consideration and look forward to your decision regarding our revised manuscript.

Best regards,

Yongming Sun

~~~~~

Yongming Sun, Ph.D., Professor

Wuhan National Laboratory for Optoelectronics

Huazhong University of Science and Technology

Wuhan 430074, China

Email: [yongmingsun@hust.edu.cn](mailto:yongmingsun@hust.edu.cn)

## **Reviewer #1**

Comments:

The work describes a reaction-passivation procedure for the recycling of aluminum foil from battery cathodes. While the work presents an interesting approach, the work seems to be too specific to be of broad significance or impact to Nature Communications. Aluminum, while a valuable commodity, is not a critical element compared to other battery-material components. Furthermore, there are key unanswered questions in the work, especially regarding the mechanism. As such, the work is not recommended for publication in Nature Communications. Some more specific comments are given below:

**Response:** Thank you very much for your positive comment “*The work describes a reaction-passivation procedure for the recycling of aluminum foil from battery cathodes...the work presents an interesting approach...*” Also, we thank the reviewer very much for his/her critical comments and those specific ones below. During the revision, a series of additional experiments and discussions have been performed to provide more evidences to support our results and discussion. Please also see the detailed replies to the following comments.

### ***Impact and novelty of our work***

Recycling of spent lithium-ion batteries is urgent with respect to resource and economy, especially for the cathodes that contains valuable metals (such as lithium, nickel and cobalt, etc). **Among all the operations for cathode recycling, facile and efficient separation of active material layer from Al foil becomes one of the most important procedures, after which the subsequent recycling of Al foil and active material can be readily facilitated. The state-of-the-art strategies for Al foil-active material layer separation** (e.g., high-temperature treatment, dissolution of binder in cathodes using organic solvents and dissolution of Al foil with mineral acids) **face issues of high energy consumption, inevitably hazardous/greenhouse gas emission, use of abundant chemicals and complex operations. Also, there lacks efficient approach to separated active materials with well-maintained structure and low Al impurity for further energy-saving, green, direct regeneration operations.** Therefore, it is highly desirable but challenging to develop facile approaches for efficient, environment-friendly separation of active material layer and Al foil for cathode regeneration.

In this contribution:

(1) **A novel reaction-passivation driven mechanism was first proposed for easy separation of Al foil and active material layer using PA solution.** Ultrathin and dense Al-PA layer was *in-situ* formed on Al foil immediately after its wetting with PA, which terminated their further reaction and continuous Al corrosion. The integrated effect of the de-bonding driven contact failure and bubbling enabled the fast Al foil-active material layer separation.

(2) To verify the efficiency of the as-proposed separation mechanism, **we first showed complete separation of Al foil and cathode active material layer for an entire spent 102 Ah-level  $\text{LiNi}_{0.55}\text{Co}_{0.15}\text{Mn}_{0.3}\text{O}_2$  (Ni55)//graphite cell from electric vehicle within 5 mins after its immersion in PA solution.** An impressive separation rate (< 5 mins) and separation efficiency (> 99.9 %) for Al foil and Ni55 layer was realized and meanwhile the dissolved transitional metal from the Ni55 was negligible (e.g., 0.27 % for Ni, 0.08 % for Co, and 0.17 % for Mn). Intact Al foil with length of 11.5 m and pieces of Ni55 layer with several centimeters in size were obtained. Besides, the PA solution showed good cyclability.

(3) **The separated Ni55 with low Al impurity and negligible damage in material structure provided the great bonus for facile regeneration of degraded Ni55 via direct annealing** with the addition of Li salts, where the regenerated Ni55 delivered a high initial capacity of 161 mAh  $\text{g}^{-1}$  at 0.3 C and a high capacity retention of 94 % for 100 cycles, which was comparable to the stat-of-the-art fresh Ni55 cathode material.

Theoretically, this work reports a novel reaction-passivation mechanism driven separation of Al foil and active material layer of cathodes for battery recycling. Technically, we show facile and fast separation of Al foil and Ni55 layer in cathode (in 5 mins) with ultrahigh separation efficiency (>99.9%) for a practical spent cell from electric vehicles, as well as intact Al foil and Ni55 with well-maintained structure after their separation. Through the life cycle assessment (LCA) and techno-economic analysis (TEA), we believe that this work offers significant insight into green, energy-saving, efficient battery cathode regeneration, and is an important breakthrough in recycling of spent lithium-ion batteries and attracts broad interest of the community.

Regarding to the reviewer's comment "Aluminum, while a valuable commodity, is not a critical element compared to other battery-material components.", we further explain the main achievement and impact of our work as follows:

Among all the operations for cathode recycling (especially for cathode active materials), facile and efficient separation of active material layer from Al foil becomes one of the most important procedures. **This work reports a novel reaction-passivation mechanism driven separation of Al foil and active material layer of cathodes for battery recycling, after which the subsequent recycling of active material can be readily facilitated.** As a demonstration, we show that such an effective materials separation approach facilitated the direct regeneration of cathode active materials in the manuscript instead of Al foil (See Page 8, Line 213-234 and Page 9, Line 235-236 and Figure 4, Page 21).

*“Figure S28 showed the dissolved of Li, Ni, Co and Mn contents from the degraded Ni55 after its separation with Al foil in PA solution, and the results indicated that the dissolution of these elements was negligible (e.g., 0.27% for Ni, 0.08% for Co, and 0.17% for Mn). Separated Ni55 with well-maintained structure and composition provided the basis for facile regeneration via direct annealing of its hybrid with Li salts (Figure 2f and Table S1). Ni55 cathode from spent cell before Al foil-active material layer separation operation was denoted as degraded Ni55 for simplification. As shown in Figure S29-30, scanning electron microscope (SEM) result revealed that the cracks in the degraded Ni55 particles cured after their recovery. Besides, TEM and XRD results verified the pure phase of layered  $\alpha$ -NaFeO2 structure with R-3m space group for the regenerated Ni55 (Figure 4a-b and S31-32). Its (108) and (110) peaks moved towards each other and (101) peak shifted to a lower degree, suggesting the transformation from cation disorder to order arrangement and thus successful materials recovery (Figure S33). As shown in Figure 4c, the regenerated Ni55 displayed much higher discharge capacities than that before regeneration (166 vs. 152 mAh g-1 for Ni55 before and after the regeneration). Moreover, the regenerated Ni55 delivered high discharge capacities of 161, 157, 146 and 132 mAh g-1 at 0.3, 0.5, 1.0 and 2.0 C, respectively, far outperforming the counterpart before regeneration (e.g., 91 mAh g-1 at 2.0 C, Figure. 4d). Also, high capacity retention of 94 % was realized after 100 cycles for the regenerated Ni55 at 0.3 C (Figure 4e). The successful regeneration of the degraded Ni55 was also evidenced by the results of EIS, constant-current charge/discharge and cyclic voltammetry test (Figure S34-38 and Table S3-4).”*

The importance and broad impact of this work have been described as follows (**See Page 3, Line 68-71, Page 10, Line 291-292 and Page 11, Line 293-294**):

*“Therefore, it is highly desirable but challenging to develop facile approaches based new mechanism for efficient and environment-friendly separation of active material and Al foil with low energy consumption.”*

*“Our work provides a promising route for facile separation of metallic current collector and active material layer, which is very different from the traditional approaches, and could promote green and energy-saving battery recycling towards practical applications.”*

To show significance and impact of this work more clearly, environment and economic analysis of PA involved direct recycling approach has been performed, and the corresponding discussion and figure have been added in the revised manuscript (**See Page 9-10**):

### ***Environmental and economic analysis of PA involved regeneration routine***

*Procedures for different recycling approaches were schematically shown, including PA involved direct recycling (PA-direct, Figure 5a and S39), general direct recycling (General-direct, Figure S40), pyrometallurgical recycling (Pyro, Figure S41) and hydrometallurgical recycling (Hydro, Figure S42). The EverBatt model developed by the Argonne National Laboratory was used for the life cycle assessment (LCA) and techno-economic analysis (TEA) of the above recycling processes based on the treatment of 10,000 tons of spent Ni55 cells (Table S5).35 Without the need of high energy consumption for cathode pre-treatment in the industrial recycling approaches (e.g. shredding, milling/thermal treatment, and sieving), the total energy consumption of the PA-direct was 5.84 MJ kg-1 cell (4.17 and 1.67 MJ kg-1 corresponding to materials use and processing, respectively), which was much lower than the General-direct, Hydro and Pyro approaches (Figure 5b and Table S6). Meanwhile, the additional GHG emission for burning mixed graphite and smelting Al and Cu scraps were avoided in comparison to other recycling approaches due to the advantage of complete separation of cathode from other battery components in our PA-direct (Figure S39 and 43). Thus, the lowest GHG emission of 0.61 g kg-1 was achieved for PA-direct (Figure 5c and Table S7). PA solution was recyclable, which enable low consumption of water. Only 2.65 L kg-1 of water was needed for treating one kilogram of cells, which was comparable to the water consumption of the Pyro process (1.97 L kg-1) and much lower than those of the Hydro (9.58 L kg-1) and General-direct (14.17 L kg-1) processes (Figure 5d and Table S8).*

*Figure 5e showed the costs of the above different recycling approaches. The use of Li salt for cathode material repair with the PA-direct (from the degraded Ni55 to regenerated Ni55) led to a slightly high cost of 6.78 \$ kg-1 (Table S9-12). PA-direct possessed the advantage of high separation efficiency, production of cathode material and direct output of high-performance regenerated cathode material (Figure S44 and Table S13). Thus, it brought higher revenue and net profit (15.79 and 9.01 \$ kg-1, respectively) than other recycling approaches (Figure 5f-g and Table S5 and 14). We performed the TEA based on manufacturing 1 kg-Ni55 cathode as a reference for industrial production process. Figure 5h-i and Table S15 showed the cost and profit for manufacturing 1 kg-Ni55 cathode from raw materials (Ni salt, Co salt, Mn salt and Li salt) and recycled materials (degraded Ni55). It is noted that more Li salt was needed for the re-synthesis of active cathode materials using the products in Pro and Hydro as raw materials. The cost for Ni55 cathode manufacture via PA-direct was only 16.07 \$ kg-1, much lower than 26.41\$ kg-1 for synthesis with raw materials (Virgin) and other processes (17.52 \$ kg-1 for General-direct, 24.32 \$ kg-1 for Pyro, 18.73 \$ kg-1 for Hydro, respectively). Thus, the PA-direct for Ni55 manufacture could achieve a high profit of 16.80 \$ kg-1, which was ~2.68 times higher than manufacture from raw materials. As a result, our PA-direct provides a promising route for facile separation of Al foil and active material layer, and active materials regeneration, which can*

*promote energy-saving, environmentally friendly and high-value battery recycling towards practical applications.*

**Figure 5. Economic and environmental analysis of PA-direct and other recycling approaches.** (a) Brief schematic of the PA-direct. (b) Energy consumption, (c) GHG emission, (d) water consumption, (e) cost, (f) revenue and (g) profit for PA-direct, General-direct, Pyro and Hydro. (h) The overall cost of manufacturing 1 kg-Ni55 cathode from raw and recycled materials. (i) Comprehensive comparison of different recycling approaches. (See Page 22, Manuscript)

1. The formation mechanism and the status of the Al-PA layer should be clearer and more supported experimentally. Can the authors demonstrate the existence of the Al-PA layer on Al foil and their morphology clearly by TEM-EDS mapping?

**Response:** We acknowledge the reviewer's careful comment and helpful suggestion.

The formation mechanism of Al-PA layer was described as follows (Page 3, Line 75-87 manuscript): "...the strong acidity of PA can induce its fast reaction with surficial Al2O3 and

*metallic Al on Al foil to produce Al3+ ions (eq. 1)...The PA molecule would immediately chelate with Al3+ to form Al-PA complex (Al-PA) (eq. 2) and terminate the further corrosion reaction between PA and surficial Al.”* The existence of PO43-, phytate and HPO42- (consisting of P-O bonding) on S-Al foil was experimentally verified by the results of Fourier transform infrared spectroscopy (FTIR) measurement (Figure 3b), which was ascribed to PA in Al-PA layer. The results of X-ray photoelectron spectroscopy (XPS) measurement suggested the existence of O, P and Al elements on Al foil surface, and the bonding between O, P and Al (Figure S18). Thus, the formation of Al-PA layer was confirmed on the surface of S-Al foil.

As suggested, the Al-PA layer on Al was investigated with TEM. An amorphous surface layer of ~20 nm was observed on the surface of Al (Figure S20a-b). The existence of P, O, C and Al elements was confirmed by TEM-EDS mapping, which, together with the fourier transform infrared spectroscopy (FTIR) results (Figure 3b), suggested the formation of the Al-PA layer (Figure S20c-d). Correspondingly, the element content from the surface to inner was also plotted and ~20 nm in thickness was evidenced for the as-formed Al-PA layer.

We further conducted the measurement of Auger electron spectroscopy to reveal the elemental signal on the surface layer of Al foil. The existence of P, O and C elements was again verified, and all the signals were gradually reduced upon Ar+ sputtering and disappeared after 160 s. Thickness of ~20 nm for the Al-PA layer was calculated based on silicon oxide wafer reference.

The following statements have been added in the revised manuscript (**See Page 6, Line 160-163**):

*“The formation of dense Al-PA layer was also verified by the results of Auger electron spectroscopy (AES), high-resolution transmission electron microscope (HRTEM) and the corresponding energy dispersive spectrometer (TEM-EDS) mapping measurements (Figure S19-20).”*

The following statements and figures have been added in the supplementary information (**See Figure S19-20, Page 9-10**):

**Figure S19.** AES for Al-PA layer on Al foil with different  $\text{Ar}^+$  sputtering times. (a) Direct spectra and (b-d) high-resolution spectra of (b) C, (c) O and (d) Al elements, and (e) the differential spectra of (a) for S-Al foil after different  $\text{Ar}^+$  sputtering times. The existence of P, O and C elements was again verified, and all the signals were gradually reduced upon  $\text{Ar}^+$  sputtering and disappeared after 160 s. Thickness of  $\sim 20$  nm for the Al-PA layer was calculated based on silicon oxide wafer reference.

**Figure S20.** (a) High-resolution transmission electron microscope (HRTEM) image of Al-PA layer. (b) Dark-field TEM image and the corresponding (c) energy dispersive spectrometer (TEM-EDS) mapping images for (d) P, O, C and Al elements for Al-PA layer on Al. An amorphous surface layer of  $\sim 20$  nm was observed on the surface of Al (Figure S20a-b). The existence of P, O, C and Al elements was confirmed by TEM-EDS mapping, which, together with the Fourier transform infrared spectroscopy (FTIR) results (Figure 3b), suggested the formation of dense Al-PA layer (Figure S20c-d). Correspondingly, the element content from the surface to inner was also plotted and  $\sim 20$  nm in thickness was shown for the as-formed Al-PA layer.

2. Separation Time dependent Al-PA thickness changes can be good evidence for the Al-PA layer formation.

**Response:** Thank you very much for this careful comment.

The thickness of Al-PA layer on S-Al foil with PA treatment time of 5 mins was analyzed by XPS depth detection investigation on P-element content changes upon Ar+ sputtering (Figure 3c). During the Ar+ sputtering, the intensity for P-O peak in high-resolution P 2s spectrum decreased gradually from 0 to 90 s, remained stable from 90 to 120 s, and disappeared after 120 s. The thickness of Al-PA layer was then estimated as ~20 nm. We further conducted the XPS measurements for the S-Al foils with PA treatment times of 30 and 60 mins. The signal for P-O peak in high-resolution P 2s spectrum disappeared after 120 s for both samples, which was consistent with the results for the S-Al foil with PA treatment time of 5 mins. Therefore, the thickness of the Al-PA layer remained constant after 5 mins-PA treatment, suggesting the formation of stable and dense Al-PA layer and successful passivation of Al foil.

The following statements have been added in the manuscript (See Page 7, Line 184-188):

*“This result supported the quick formation of dense Al-PA layer on Al foil surface, which inhibited the continuous Al dissolution. The formation of stable, ultrathin Al-PA layer was further evidenced by the results of XPS investigation on Al foils with different PA treatment times (Figure S21).”*

The following statement and figures have been added in the supplementary information (See Figure S21, Page 10):

**Figure S21.** High-resolution P 2s XPS spectra of S-Al foils with PA treatment times of (a) 30 and (b) 60 mins. The signal for P-O peak in high-resolution P 2s spectrum disappeared after 120 s for both samples, consistent with the results for the S-Al foil with PA treatment time of 5 mins.

3. Miller indices should be displayed in the XRD patterns since the authors discussed them on p.5

**Response:** Thank you very much for your careful comment.

As suggested, the miller indices were added in the XRD patterns in Figure 2f, 4b, S16-17. (See Figure 2f and 4b, Page 19 and 21 in the revised manuscript, see Figure S16-17, Page 8 in the supplementary information).

**Figure 2.** (f) XRD patterns of the separated Al foil and Ni55 layer.

**Figure 4.** (b) XRD pattern of the regenerated Ni55.

**Figure S16 (Figure S15 in the initial supplementary information).** XRD patterns of the Ni55 separated by using HCl solution.

**Figure S17 (Figure S16 in the initial supplementary information).** XRD patterns of F-Al foil and degraded Ni55.

4. P7. Line 196, the sentence "XRD result confirmed the highly ordered layered structure" is vague. Should mention the crystal structure instead of the "ordered layered structure"

**Response:** We acknowledge the reviewer very much for this helpful suggestion.

We carefully described the crystal structure in the revised manuscript as follows (See Page 8, Line 220-227).

The initial statement "Scanning electron microscope (SEM) result revealed that the cracks in the degraded Ni55 particles cured after their recovery, and XRD result confirmed the highly ordered layered structure of the regenerated Ni55 (Figure 4a-b and S20-23), supporting the well recovered structure of Ni55." has been replaced as "As shown in Figure S29-30, scanning electron microscope (SEM) result revealed that the cracks in the degraded Ni55 particles cured after their recovery. Besides, HRTEM and XRD results verified the pure phase of layered  $\alpha$ -NaFeO2 structure with R-3m space group for the regenerated Ni55 (Figure 4a-b and S31-32).34 Its (108) and (110) peaks moved towards each other and (101) peak shifted to a lower degree, supporting the transformation from cation disorder to order arrangement and thus successful materials recovery (Figure S33).35"

#### References

34 J.X. Wang, K. Jia, J. Ma, Z. Liang, Z.F. Zhuang, Y. Zhao, B.H. Li, G.M. Zhou, H.M. Cheng, *Nat. Sustain.* 2023, 10.1038/s41893-023-01094-9.

35 T. Wang, H. M. Luo, J. T. Fan, B. P. Thapaliya, Y. C. Bai, I. Belharouak and S. Dai, *iScience*, 2022, 25, 103801.

We have added the following figure in the supplementary information (See Figure S33, Page 14).

**Figure S33.** Zoom-in XRD peaks of degraded and regenerated Ni55.

5. As a control, the authors can polish out the Al-PA layer after the separation and monitor the Al content change in PA solution again to make sure the role of Al-PA layer.

**Response:** We thank the reviewer very much for this thoughtful comment.

As suggested, the Al-PA layer was polished by 3000 grit sandpaper, and then was rested in a fresh PA solution under the same condition as the initial treatment. According to the ICP-MS measurement, the remaining PA solution showed Al3+ concentration of 0.472 wt%, and the value was very close to 0.478 wt% for the used PA solution for the active material-Al foil separation. Thus, passivation process took place again for the polished Al, and metallic Al foil could be well protected by the as-formed stable PA-layer.

The following statements have been added in the manuscript (See Page 7, Line 188-192):

*“To verify the capability of producing stable Al-PA layer, the as-formed Al-PA layer was polished and then treated with fresh PA solution under the same condition as the initial treatment. This PA solution showed similar Al3+ concentration to that for the initial Al foil-active materials layer separation (0.472 wt% and 0.478 wt%, respectively).”*

6. The authors should confirm that PA react with the other active materials and form the passivation layer.

**Response:** We would like to acknowledge for your careful comment.

To show the generality of reaction-passivation mechanism driven Al foil-active material layer separation, we further performed the experiments on spent LCO cathodes. The results showed that the LCO layer was completely separated from the Al foil in PA solution in 5 mins. The obtained Al foil demonstrated a clean surface without any residual active material or observed damage (Figure S22). Besides, the thickness of the Al foil remained unchanged before and after separation (Figure S23). More elaborate measurement was further subject to ICP-MS measurement. The results indicated a low dissolution ratio (1.68 wt%) for LCO cathode in the used PA solution (Figure S24). The low Al content in the used PA solution and stable S-Al foil supported the formation of passivation layer on its surface. XRD measurement was further conducted to investigate the separated LCO materials. The separated LCO showed same XRD peaks as that before the PA treatment (Figure S25). Therefore, the above results verified that Al foil could be passivated instead of continuous corrosion after the initial reaction for Al foil-active material layer separation, where the similar results were also observed for spent LFP cathodes (Figure S26-27). We have added the above statements and following figures in the supplementary information (See Figure S22-27, Page 11-12).

**Figure S22.** The digital images for Al foil and active material layer separation of LCO cathode in 30 wt% PA solution.

**Figure S23.** SEM images for (a) the LCO cathode and (b) the S-Al foil.

**Figure S24.** Al, Co and Li contents in PA solution after Al foil-LCO layer separation by ICP-MS measurement.

**Figure S25.** XRD patterns of LCO before and after Al foil-LCO layer separation.

**Figure S26.** The digital images for Al foil and active material layer separation of LFP cathode in 30 wt% PA solution.

**Figure S27.** SEM images for (a) the LFP cathode and (b) S-Al foil.

The following statements have been added in the manuscript (See Page 8, Line 207-211):

*“Such active material layer-Al foil separation approach was also employed for the separation of Al foil and active material layer in other cathodes including  $\text{LiCoO}_2$  and  $\text{LiFePO}_4$  (Figure S22-27), and facile separation was realized for both the cathodes with high separation efficiency and low PA consumption, which further supported the importance and impact of our PA-direct approach.”*

**Reviewer #2**

Comments:

Recycling LIBs in an important topic and the method shown is a novel approach although there are many similar organic acids which have been used. The idea of selective passivation of metals during dissolution is known in metal processing although it has not been applied to LIB batteries. The approach clearly works although there are two obvious issues which the authors have not addressed.

**Response:** Thank you very much for your positive comment *“Recycling LIBs in an important topic and the method shown is a novel approach ... The approach clearly works...”*. During the revision, a series of additional calculations and discussions about LCA/TEA have been

performed to provide more evidences to support our results and the impacts of our work. Please also see the detailed replies to the following comments.

1. PA even on large scale bulk purchase is >\$50/kg and since it is a stoichiometric reagent and used up in the delamination process will make the process economics unviable.

**Response:** Thank you very much for your valuable comment and suggestion. To address the reviewer's concern about the economics for using PA, we collected the price information of PA solution (50 wt%) from 10 companies in China (Table S10), and the average of which was ~5.76 \$ kg-1 for large-scale bulk purchase. The calculated theoretic consumption of PA was as low as ~48.75 g for the treatment of 1 kg-spent cell according to the amount of dissolved Al in PA solution and Al-PA layer on Al foil, and the used PA solution showed good recyclability with high separation efficiency > 99.9% (Figure 3g). Therefore, the PA cost for the treatment of degraded cathodes in 1 kg-cell could be as low as 0.56 \$ kg-1, supporting that using PA is potentially economic and feasible for scalability.

The following Table and statement have been added in the supplementary information (**See Table S10, Page 22**):

**Table S10.** Prices for 50 wt% PA solution from different companies. The price information of PA solution (50 wt%) from 10 companies was collected in May 2023, and the average of which was ~5.76 \$ kg-1 for large-scale bulk purchase.

| Website                                                                                                     | Price (\$ kg -1 ) |
|-------------------------------------------------------------------------------------------------------------|------------------------------|
| http://www.sbczh.com/tf_product.asp?ln=0             | 5.81                         |
| http://gzdshg168.com/                                                   | 5.81                         |
| http://61819185013.cn.gongxuku.com/credit/         | 8.34                         |
| https://wap.21food.cn/company/info1417749.html | 5.81                         |
| http://www.jscwskj.com/                                               | 5.29                         |
| https://gzsanchanghg.company.lookchem.cn/           | 5.08                         |
| http://www.zzfthg.com/                                                 | 7.55                         |
| https://sdacswkj.cn.china.cn/                                   | 3.78                         |
| http://www.condicechem.com/                                       | 3.63                         |
| http://www.hongtaobio.com/                                         | 6.54                         |
| Average price                                                                                               | 5.76                         |

**Figure 3.** (g) The separation efficiency vs. cycle number plot for Al foil-Ni55 layer separation with repeated use of the same PA solution.

2. There are faster delamination processes reported which do not have stoichiometric reagents e.g. *Green Chem.*, 2021, 23, 4710-4715

**Response:** Thank you very much for this careful comment. We carefully compared the as-mentioned reference and highlighted the advantage of our reaction-passivation mechanism driven materials separation as follows.

- 1) **Separation mechanism.** We first proposed a reaction-passivation driven mechanism for facile separation of Al foil and active material layer, which differed from the reported approaches including acid leaching, flotation, and mechanical processing, scientifically (e.g., sonication in the mentioned *Green Chemistry* paper).
- 2) **Separation efficiency.** We highlighted the merit of high separation efficiency for the reaction-passivation driven mechanism. Once the cathode was immersed into the PA solution, Al foil and active material layer would be divided rapidly, which did not depend on the total amount of the treated cathode. High separation efficiency of above 99.9% was achieved for Al foil-Ni55 active material layer separation for a batch of cathode with total area of 9200 cm2 from a spent 102 Ah cell in 5 mins, demonstrating a recorded-high separation efficiency for kg-level Al foil-active material layer separation.
- 3) **Energy consumption.** Our separation approach did not need to involve additional energy-consumption process during the separation operation, such as thermal annealing, mechanical sieving and sonication. For example, high power of 120 W cm-2 was employed for the ultrasonication separation of a small piece of cathode in the mentioned *Green Chemistry* paper.
- 4) **Postprocessing.** Meter-level Al foil could be easily separated from the active material layer due to its good integrity, supporting facile postprocessing. In contrast, much more complex operation was required for the Al foil and active materials separation for the other reported separation approaches. For example, sonication in the mentioned *Green Chemistry* paper could bring in the cracks and pulverization of Al foil, leading to difficulty for Al foil-active material layer separation with high quality and high efficiency. In fact, mechanical separation is often less precise (*Journal of Cleaner Production* 2022, 340,130535). We emphasized that

the Al residues in the separated Ni55 cathode were very low (0.026 wt%), which provides great basis for materials regeneration.

5) **Potential scalability.** The as-proposed separation approach possesses the advantage of easy operation and does not involve complex equipment, and thus support scalable active material-Al foil separation. As a prototype, we successfully demonstrated Al foil-Ni55 active material layer separation with a total electrode area of 9200 cm2 and mass of ~705 g from a spent 102 Ah cell in 5 mins with simply immersing the cathode into PA solution. In contrast, all the previous publications only demonstrate the Al foil-active material layer separation with only several pieces of cathodes, such as ~ 440 cm2 (19.5 cm × 22.5 cm) in the mentioned *Green Chemistry* paper.

3. The process does still not remove the polymer binder. The major contributions to the LCA is the thermal treatment to remove the binder and the cost of the lixiviant to delaminate the electrode. Since PA is competing against sulfuric acid at \$0.4 /kg it will never compete.

**Response:** We acknowledge the reviewer's careful comments.

***Polymer binder is indeed not removed during our Al foil-cathode active materials separation process, but is removed during the subsequent post treatment process accompanied by the direct active cathode materials regeneration. Thus, no additional energy consumption for binder removal during traditional recycling of cathode materials is required, suggesting the potential advantage of energy saving.***

LCA is the assessment of all environmental burdens regarding a product, including consumption of energy, greenhouse gas emissions and solid waste generation for environment impacts. Thus, when evaluating the LCA of battery materials recycling, one needs to consider the energy consumption during the electrode processing (e.g., separation of active materials and current collector) and the thermal treatment for removing the binder and conductive additive in cathode active material layer, and the carbon-based anode active materials. Besides, the generation of greenhouse gas during thermal annealing is another important parameter. ***Our Al foil-active material layer separation does not involve high energy consumption for the pre-treatment of cathode in current industrial processing, such as shredding, milling/thermal treatment, and sieving. The binder is removed accompanied by the direct regeneration process of the active cathode material, without the consumption of additional energy as the traditional recycling of cathode materials.*** Overall, major contributions to the LCA are the pre-thermal and post-thermal treatment to remove the binder and graphite as well as smelting Al and Cu residues. Our Al foil-cathode active materials separation approach gets high marks regarding to these aspects.

To address the reviewer's concern about the economics for using PA, we collected the price information of PA solution (50 wt%) from 10 companies in China (Table S10), the average of which was ~5.76 \$ kg-1 for large scale bulk purchase. The calculated theoretic consumption of

PA was as low as ~48.75 g for the treatment of 1 kg-spent cell according to the amount of dissolved Al in PA solution and Al-PA layer on Al foil, and the used PA solution showed good recyclability with high separation efficiency > 99.9% (Figure 3g). The PA cost for the treatment of 1 kg-spent cell could be as low as 0.56 \$ kg-1, which suggested that using PA is potentially economic and feasible for scalability.

During the traditional battery recycling process (e.g., pyrometallurgy and hydrometallurgy), lixiviant was used to delaminate the electrode. Taking sulfuric acid as an example, the theoretical consumption of lixiviant is about 943 g for leaching degraded cathodes in 1 kg-cell through the following equipment:

Therefore, the cost for using PA is only slightly higher than using sulfuric acid for electrode treatment. When one considers the energy consumption and complex operations in traditional recycling processes, we believe that our cathode material separation and regeneration with PA can far outperform the traditional active materials recycling (e.g., pyrometallurgy and hydrometallurgy) with lixiviant for material separation.

**Figure 3.** (g) The separation efficiency vs. cycle number plot for Al foil-Ni55 layer separation with repeated use of the same PA solution.

4. It is a nice paper but the impact is not high enough for Nature Communications. It really needs LCA/TEA and a comparison with competing techniques to increase the impact.

**Response:** Thank you very much for your positive comment “It is a nice paper...”. As suggested, we conducted the LCA/TEA and made a comparison with competing techniques to increase the impact of our work.

### Environmental and economic analysis of PA based regeneration routine

Procedures for different recycling approaches were schematically shown, including PA involved direct recycling (PA-direct, Figure 5a and S39), general direct recycling (General-direct, Figure S40), pyrometallurgical recycling (Pyro, Figure S41) and hydrometallurgical recycling (Hydro,

Figure S42). For example, during a General-direct process, all components of the spent cell are crushed, pre-thermal treatment, floatation and degraded Ni55 is ultimately repaired via an annealing process (Figure S39). In this section, the EverBatt model developed by the Argonne National Laboratory was used to analyze the LCA and TEA for above recycling process based on the treatment of 10,000 tons of spent Ni55 cells (Table S5).

LCA: (1) **Energy consumption**. Our PA-direct does not involve high energy consumption for the pre-treatment of cathode in the mentioned industrial recycling processes (General-direct, Pyro and Hydro), such as shredding, milling/thermal treatment, and sieving. The binder is removed accompanied by the direct regeneration process of the active cathode material, without the consumption of additional energy like traditional recycling processing of cathode materials. Low energy consumption of 5.84 MJ kg-1 is finally achieved (Figure 5b and Table S6). (2) **GHG emission**. Meanwhile, the additional GHG emission for burning mixed graphite and smelting Al and Cu scraps are avoided in comparison to other recycling approaches due to the advantage of complete separation of cathode from other battery components in our PA-direct (Figure S39 and 43). Thus, the lowest GHG emission of 0.61 g kg-1 is achieved (Figure 5c and Table S7). (3) **Water consumption**. Since PA solution can be reused, only 2.65 L kg-1 of water is needed for treating one kilogram of cells, which is comparable to the water consumption of the Pyro process (1.97 L kg-1) and much lower than those of the Hydro (9.58 L kg-1) and General-direct (14.17 L kg-1) processes (Figure 5d and Table S8).

TEA: (1) **Cost for processing**. The overall cost for labour and others is similar for different recycling approaches. Li salt is needed in the PA-direct for the repairing of cathode material (from the degraded Ni55 to regenerated Ni55), leading to relatively high cost of 6.78 \$ kg-1 (Figure 5e and Table S9-12). It is noted that more Li salt is needed for the re-synthesis of active cathode materials using the product in Pro and Hydro as raw materials, which will be discussed later. (2) **Profit**. PA-direct possesses the advantage of high separation efficiency and production (>99.9 %) of cathode material and direct output of high-performance regenerated cathode material (Figure S44 and Table S13). Thus, it brings higher revenue and net profit of 15.79 and 9.01 \$ kg-1 in comparison to other recycling approaches (Figure 5f and g and Table S5 and 14) **Overall cost**. We performed the TEA based on manufacturing 1 kg-Ni55 cathode as a reference for industrial production process. Figure 5h and Table S15 shows the cost and potential profit for manufacturing 1 kg-Ni55 cathode from raw materials (Ni salt, Co salt, Mn salt and Li salt) and recycled materials (degraded Ni55). The cost for Ni55 cathode produced via PA-direct is only 16.07 \$ kg-1, much lower than manufacturing from raw materials (Virgin, 26.41\$ kg-1) and other process (17.52 \$ kg-1 for General-direct, 24.32 \$ kg-1 for Pyro, 18.73 \$ kg-1 for Hydro). The PA involved regeneration process for Ni55 fabrication achieves a high profit of 16.80 \$ kg-1, which is ~2.68 times higher than manufacturing from raw materials.

As a result, our PA-direct provides a promising route for facile separation of metallic current collector and active material layer, and active materials regeneration, which can promote energy-saving, environmentally friendly and high value towards practical applications (Figure 5i).

The following section have been added in the manuscript (See Page 9-10):

### ***Environmental and economic analysis of PA involved regeneration routine***

*Procedures for different recycling approaches were schematically shown, including PA involved direct recycling (PA-direct, Figure 5a and S39), general direct recycling (General-direct, Figure S40), pyrometallurgical recycling (Pyro, Figure S41) and hydrometallurgical recycling (Hydro, Figure S42). The EverBatt model developed by the Argonne National Laboratory was used for the life cycle assessment (LCA) and techno-economic analysis (TEA) of the above recycling processes based on the treatment of 10,000 tons of spent Ni55 cells (Table S5).35 Without the need of high energy consumption for cathode pre-treatment in the industrial recycling approaches (e.g. shredding, milling/thermal treatment, and sieving), the total energy consumption of the PA-direct was 5.84 MJ kg-1 cell (4.17 and 1.67 MJ kg-1 corresponding to materials use and processing, respectively), which was much lower than the General-direct, Hydro and Pyro approaches (Figure 5b and Table S6). Meanwhile, the additional GHG emission for burning mixed graphite and smelting Al and Cu scraps were avoided in comparison to other recycling approaches due to the advantage of complete separation of cathode from other battery components in our PA-direct (Figure S39 and 43). Thus, the lowest GHG emission of 0.61 g kg-1 was achieved for PA-direct (Figure 5c and Table S7). PA solution was recyclable, which enable low consumption of water. Only 2.65 L kg-1 of water was needed for treating one kilogram of cells, which was comparable to the water consumption of the Pyro process (1.97 L kg-1) and much lower than those of the Hydro (9.58 L kg-1) and General-direct (14.17 L kg-1) processes (Figure 5d and Table S8).*

*Figure 5e showed the costs of the above different recycling approaches. The use of Li salt for cathode material repair with the PA-direct (from the degraded Ni55 to regenerated Ni55) led to a slightly high cost of 6.78 \$ kg-1 (Table S9-12). PA-direct possessed the advantage of high separation efficiency, production of cathode material and direct output of high-performance regenerated cathode material (Figure S44 and Table S13). Thus, it brought higher revenue and net profit (15.79 and 9.01 \$ kg-1, respectively) than other recycling approaches (Figure 5f-g and Table S5 and 14). We performed the TEA based on manufacturing 1 kg-Ni55 cathode as a reference for industrial production process. Figure 5h-i and Table S15 showed the cost and profit for manufacturing 1 kg-Ni55 cathode from raw materials (Ni salt, Co salt, Mn salt and Li salt) and recycled materials (degraded Ni55). It is noted that more Li salt was needed for the re-synthesis of active cathode materials using the products in Pro and Hydro as raw materials. The cost for Ni55 cathode manufacture via PA-direct was only 16.07 \$ kg-1, much lower than 26.41\$*

$\text{kg}^{-1}$  for synthesis with raw materials (Virgin) and other processes (17.52  $\text{\$ kg}^{-1}$  for General-direct, 24.32  $\text{\$ kg}^{-1}$  for Pyro, 18.73  $\text{\$ kg}^{-1}$  for Hydro, respectively). Thus, the PA-direct for Ni55 manufacture could achieve a high profit of 16.80  $\text{\$ kg}^{-1}$ , which was  $\sim 2.68$  times higher than manufacture from raw materials. As a result, our PA-direct provides a promising route for facile separation of Al foil and active material layer, and active materials regeneration, which can promote energy-saving, environmentally friendly and high-value battery recycling towards practical applications.

The following figures have been added in the manuscript (See Page 22):

**Figure 5.** Economic and environmental analysis of PA-direct and other recycling approaches. (a) Brief schematic of the PA-direct. (b) Energy consumption, (c) GHG emission, (d) water consumption, (e) cost, (f) revenue and (g) profit for PA-direct, General-direct, Pyro and Hydro. (h) The overall cost of manufacturing 1 kg-Ni55 cathode from raw and recycled materials. (i) Comprehensive comparison of different recycling processes.

The following figures have been added in the supplementary information (See **Figure S39-44**, **Page 16-17**):

**Figure S39.** Process diagram of PA involved direct recycling (PA-direct).

**Figure S40.** Process diagram of general direct recycling (General-direct).

**Figure S41.** Process diagram of general pyrometallurgical recycling (Pyro).

**Figure S42.** Process diagram of general hydrometallurgical recycling (Hydro).

**Figure S43.** Digital images for (a) automatic disassembly of 102 Ah-spent Ni55 cell and (b) the obtained cathode.

**Figure S44.** Recycling efficiency of different recycling approaches.

The following tables have been added in the supplementary information (See Table S5-15, Page 20-24):

**Table S5.** LCA and TEA of different recycling approaches.

|                                             | Pyro | Hydro | General-direct | PA-direct |
|---------------------------------------------|-------------|--------------|-----------------------|------------------|
| Total energy (MJ kg -1 , cell)   |             |              |                       |                  |
| Total Energy                                | 10.71       | 19.57        | 20.92                 | 5.84             |
| Fossil fuels                                | 9.86        | 18.08        | 18.77                 | 5.58             |
| Coal                                        | 2.34        | 3.11         | 13.96                 | 2.04             |
| Natural gas                                 | 6.63        | 13.41        | 2.47                  | 1.45             |
| Petroleum                                   | 0.89        | 1.56         | 2.34                  | 2.10             |
| Water consumption (gal kg -1 )   | 0.5         | 2.5          | 3.7                   | 0.7              |
| Total Emissions (g kg -1 , cell) |             |              |                       |                  |
| VOC                                         | 0.12        | 0.21         | 0.32                  | 0.21             |
| CO                                          | 0.43        | 0.75         | 0.89                  | 0.55             |
| NO x                             | 0.89        | 1.77         | 2.19                  | 1.51             |
| PM10                                        | 0.08        | 0.15         | 0.43                  | 0.22             |
| PM2.5                                       | 0.05        | 0.10         | 0.26                  | 0.18             |
| SO x                             | 0.75        | 22.86        | 3.52                  | 0.64             |
| BC                                          | 0.02        | 0.02         | 0.05                  | 0.05             |
| OC                                          | 0.01        | 0.03         | 0.07                  | 0.06             |
| CH 4                             | 1.18        | 2.12         | 2.54                  | 0.73             |
| N 2 O                            | 0.01        | 0.02         | 0.03                  | 0.01             |
| CO 2                             | 2143        | 1396         | 1810                  | 580              |
| CO 2 (w/C in VOC & CO)           | 2144        | 1398         | 1813                  | 582              |
| GHGs                                        | 2183        | 1468         | 1896                  | 605              |
| Revenue (\$ kg -1 , cell)        | 4.90        | 6.35         | 14.10                 | 15.79            |
| Cost (\$ kg -1 , cell)           | 4.87        | 4.35         | 6.59                  | 6.78             |
| Profit (\$ kg -1 , cell)         | 0.03        | 2.00         | 7.51                  | 9.01             |

**Table S6.** Energy consumption of different recycling approaches (MJ kg-1, cell).

|                       | Material input | Energy input | Total |
|-----------------------|----------------|--------------|-------|
| Pyro           | 7.72           | 2.99         | 10.71 |
| Hydro          | 17.45          | 2.12         | 19.57 |
| General-direct | 4.79           | 16.12        | 20.92 |
| PA-direct      | 4.17           | 1.67         | 5.84  |

**Table S7.** GHG emission of different recycling approaches (g kg-1, cell).

|                       | Material input | Energy input | Process |
|-----------------------|----------------|--------------|---------|
| Pyro           | 0.48           | 0.22         | 1.48    |
| Hydro          | 1.10           | 1.45         | 0.22    |
| General-direct | 0.48           | 1.35         | 0.07    |
| PA-direct      | 0.41           | 0.13         | 0.07    |

**Table S8.** Water consumption of different recycling approaches (L kg-1, cell).

|                       | Material input | Energy input | Process |
|-----------------------|----------------|--------------|---------|
| Pyro           | 1.29           | 0.68         | 0       |
| Hydro          | 5.57           | 0.22         | 3.79    |
| General-direct | 2.49           | 7.90         | 3.79    |
| PA-direct      | 2.14           | 0.44         | 0.07    |

**Table S9.** Materials requirements for 1 kg-spent cell recycling with different approaches.

| Pyro       | Hydro      | General-direct | PA-direct  |
|-------------------|-------------------|-----------------------|-------------------|
| Hydrochloric Acid | Sulfuric Acid     | Lithium hydroxide     | Lithium Hydroxide |
| Limestone         | Hydrogen Peroxide | Lime                  | PA solution       |
| Hydrogen Peroxide | Hydrochloric Acid | Sodium Hydroxide      |                   |
| Sand              | Soda Ash          |                       |                   |
|                   | Sodium Hydroxide  |                       |                   |

**Table S10.** Prices for 50 wt% PA solution from different companies. The price information of PA solution (50 wt%) from 10 companies was collected in May 2023, and the average of which was ~5.76 \$ kg-1 for large-scale bulk purchase.

| Website                                                                                              | Price (\$ kg-1) |
|-------------------------------------------------------------------------------------------------------------|-----------------------------------|
| http://www.scbczh.com/tf_product.asp?ln=0           | 5.81                              |
| http://gzdshg168.com/                                                   | 5.81                              |
| http://61819185013.cn.gongxuku.com/credit/         | 8.34                              |
| https://wap.21food.cn/company/info1417749.html | 5.81                              |
| http://www.jscwskj.com/                                               | 5.29                              |
| https://gzsanchanghg.company.lookchem.cn/           | 5.08                              |
| http://www.zzfthg.com/                                                 | 7.55                              |
| https://sdacswkj.cn.china.cn/                                   | 3.78                              |
| http://www.condicechem.com/                                       | 3.63                              |
| http://www.hongtaobio.com/                                         | 6.54                              |
| Average price                                                                                               | 5.76                              |

**Table S11.** Prices of materials for different cathode recycling approaches.

|                                 | Price (\$ t -1 ) | Data sources |
|---------------------------------|-----------------------------|--------------|
| PA solution (50 wt%)            | 5764.20                     | Table S10    |
| Ni55                            | 32683.54                    | 10jqka       |
| LiOH·H 2 O           | 42561.23                    | SMM          |
| Li 2 CO 3 | 32756.17                    | SMM          |
| graphite                        | 4459.487                    | SMM          |
| Al foil                         | 2396.793                    | SMM          |
| Cu foil                         | 10110.11                    | SMM          |
| HCl                             | 8.71                        | 100ppi       |
| Ammonia                         | 459.99                      | 100ppi       |
| Ammonium bicarbonate            | 196.10                      | 100ppi       |
| Hydrogen peroxide               | 115.24                      | 100ppi       |
| Ni in product                   | 23355.19                    | SMM          |
| Mn in product                   | 2742.69                     | SMM          |
| Co in product                   | 30666.03                    | SMM          |
| Water                           | 1.75958248                  | BDB          |
| Sewage treatment                | 2.20                        | 51wctt       |

**Data sources:**

SMM (<https://www.smm.cn/>), 10jqka (<https://www.10jqka.com.cn/>), 100ppi (<https://www.101ppi.com/>) and BDB (<http://sz.bendibao.com/>). The data was collected in May 2023.

**Table S12.** Cost for different recycling approaches (\$ kg-1, cell).

|                   | Pyro | Hydro | General-direct | PA-direct |
|-------------------|------|-------|----------------|-----------|
| Materials         | 0.06 | 0.26  | 2.12           | 2.11      |
| Labor             | 0.04 | 0.03  | 0.03           | 0.01      |
| Other direct cost | 0.30 | 0.17  | 0.22           | 0.26      |
| Depreciation      | 0.48 | 0.26  | 0.31           | 0.38      |
| Other fixed cost  | 0.54 | 0.30  | 0.35           | 0.42      |
| Plant overhead    | 0.14 | 0.08  | 0.09           | 0.10      |
| General expenses  | 0.24 | 0.17  | 0.42           | 0.44      |
| Battery fee       | 3.05 | 3.05  | 3.05           | 3.05      |

**Table S13.** Produced materials from different recycling approaches (kg kg-1, cell).

|               | Pyro | Hydro | General-direct | PA-direct |
|---------------|-------------|--------------|-----------------------|------------------|
| Copper        | 0.05        | 0.05         | 0.05                  | 0.06             |
| Aluminum      | NA          | 0.04         | 0.04                  | 0.04             |
| Graphite      | NA          | 0.26         | 0.26                  | 0.29             |
| Mn in product | NA          | 0.07         | NA                    | NA               |
| Ni in product | 0.13        | 0.13         | NA                    | NA               |
| Co in product | 0.05        | 0.05         | NA                    | NA               |
| Ni55 product  | NA          | NA           | 0.38                  | 0.42             |

**Table S14.** Revenue of different recycling approaches (\$ kg-1, cell).

|               | Pyro | Hydro | General-direct | PA-direct |
|---------------|-------------|--------------|-----------------------|------------------|
| Copper        | 0.55        | 0.55         | 0.55                  | 0.61             |
| Aluminum      | NA          | 0.09         | 0.09                  | 0.10             |
| Graphite      | NA          | 1.17         | 1.17                  | 1.30             |
| Mn in product | NA          | 0.19         | NA                    | NA               |
| Ni in product | 3.20        | 1.15         | NA                    | NA               |
| Co in product | 1.15        | 3.20         | NA                    | NA               |
| Ni55 product  | NA          | NA           | 12.30                 | 13.79            |

**Table S15.** The TEA of manufacturing 1 kg-cell from raw and degraded Ni55 materials.

|                       | Cost (\$ kg -1 ) |              |            | Price of Ni55 (\$ kg -1 ) | Profit (\$ kg -1 ) |
|-----------------------|-----------------------------|--------------|------------|--------------------------------------|-------------------------------|
|                       | Recycling                   | Regeneration | Production |                                      |                               |
| Virgin         | NA                          | NA           | 26.41      | 32.68                                | 6.27                          |
| Pyro           | 12.18                       | NA           | 12.14      | 32.68                                | 8.36                          |
| Hydro          | 10.87                       | NA           | 7.86       | 32.68                                | 13.95                         |
| General-direct | NA                          | 17.52        | NA         | 32.68                                | 15.35                         |
| PA-direct      | NA                          | 16.07        | NA         | 32.68                                | 16.80                         |

### **Reviewer #3**

Comments:

The rapid demand for lithium-ion batteries is widely envisaged to lead to a significant amount of battery waste. The environmental concerns along with the uncertainties in the supply chains make the development of efficient recycling technologies urgent and necessary. Due to the complex structure and inherent safety issues, spent LIBs have to be subjected to a variety of pre-treatments prior the recovery of electrode active materials through direct recycling, pyrometallurgy, hydrometallurgy, or a combination of those methods. The efficiency of the pre-treatments and especially the separation of the active material from the current collector is of paramount importance in the effort to decrease the impurities content in the recovered active materials and improve the cost and environmental credentials of the recycling process. The proposed process for materials separation through reaction-passivation of the Al current collector addresses this issue by offering a closed loop recycling process where NMC cathode materials can be easily recovered and regenerated.

PA has been widely used to develop environmentally friendly chemical conversion coatings on variety of metal alloys due to its high coordination sites and ability to chelate with many different metal ions. It was also recently applied as precipitant in hydrometallurgical recycling of  $\text{LiMn}_2\text{O}_4$  batteries. Nevertheless, the proposed utilization of PA in the pre-treatment stage is an original contribution leading to some notable advantages.

The authors present a substantial amount of experimental and analytical data demonstrating the advantages of their idea. However, the manuscript contains numerous mistakes and inconsistencies as well as deficiencies which require additional contributions. Thus, while I do not question the merits of the work, I am afraid the manuscript does not represent a sufficiently high quality to justify publication in Nature Communications. I would like to ask the Authors to consider the following comments:

**Response:** Thank you very much for your positive comment “...*The proposed process for materials separation through reaction-passivation of the Al current collector addresses this issue by offering a closed loop recycling process where NMC cathode materials can be easily recovered and regenerated...The proposed utilization of PA in the pre-treatment stage is an original contribution leading to some notable advantages. The authors present a substantial amount of experimental and analytical data demonstrating the advantages of their idea...*”. Also, we thank the reviewer very much for his/her critical comments and those specific ones below. During the revision, a series of experiments and discussions have been performed to provide more evidences to support our results and improve the quality of this manuscript. Please also see the detailed responses to the following comments.

1. In the “Introduction” section the author claims that the loss of contact between the cathode active material and the aluminium foil was due to the hydrogen gas bubbling at the interface. In support of this claim, the authors make a reference to the paper of Cao et al [18] in which graphite anodes were separated from Cu current collectors by electrolysis method, where hydrogen, continuously produced at the negative electrode, weakened the bonding force between the binder and the copper foil. The mechanism and duration of the hydrogen generation would depend on the kinetics of passivation layer formation, surface coverage etc. It is apparent that Figure 2, as presented, does not suggest continuous bubbling. Hence, the following aspect of the work is not clarified - is the gas generation at the interface the only de-bonding mechanism or could there be contribution from other de-bonding mechanisms?

**Response:** Thank you very much for your careful comments and helpful suggestions, which help us to conduct deeper analysis. Active cathode material layer was adhered to Al foil in cathode through weak van der Waals interaction between PVDF and Al foil.20 PA reacted with surficial Al2O3 and metallic Al on Al foil to produce a dense Al-PA layer with strong covalent bond interaction with Al foil,21 which, together with bubbling, could damage the interaction between PVDF and Al foil and lead to the de-bonding between them. Thus, the integrated effect of the de-bonding driven contact failure, together with bubbling, enabled the fast separation of Al foil and Ni55 layer, and explained the faster separation than Cu foil and graphite layer by only hydrogen bubbling in the reference of “Cao et al [18]”.

The following statements have been added in the revised manuscript (**See Page 3, Line 78-83**):

*“Active material layer was adhered to Al foil in cathode through weak van der Waals interaction between PVDF and Al foil.20 PA reacted with surficial Al2O3 and metallic Al on Al foil to produce a dense Al-PA layer with strong covalent bond interaction with Al foil,21 which, together with bubbling, could damage the interaction between Al foil and PVDF in active material layer.”*

References:

20 Y. Zhao, Z. Liang, Y.Q. Kang, Y.N. Zhou, Y.X. Li, X.M. He, L. Wang, W.C. Mai, X.S. Wang, G.M. Zhou, J.X. Wang, J.G. Li and N. Tavajohi and B.H. Li, *Energy Storage Mater.*, 2021, **35**, 353-377.

21 X.D. You, H. Wu, R.N. Zhang, Y.L. Su, L. Cao, Q.Q. Yu, J.Q. Yuan, K. Xiao, M.R. He, Z.Y. Jiang, *Nat. Commun.*, 2019, **10**, 4160.

2. Figure S6 presents important data on the dynamic of passivation process depending on PA concentrations. However, it is practically not discussed in the manuscript and not supported by additional characterization – e.g. Potentiodynamic polarization and/or Impedance spectroscopy of S-Al. It is well documented in the literature that at high concentration of PA the large number

of phosphoryl groups adsorbed the surface of aluminium can hinder the formation of Al-PA complexes while at low concentration the passivation film could be incomplete.

**Response:** Thank you very much for your valuable comment and suggestion.

To reveal the concentration effect of PA solution on the separation time and Al passivation, we conducted the Al foil-Ni55 layer separation in PA solutions with concentrations of 10, 20, 30, 40 and 50 wt% (namely 10 wt% PA-Al foil, 20 wt% PA-Al foil, 30 wt% PA-Al foil, 40 wt% PA-Al foil and 50 wt% PA-Al foil), respectively. The as-separated Al foils were investigated by potentiodynamic polarization and impedance spectroscopy. As showed in Figure S6-7 and Table S3, all of separated Al foil demonstrated similar small values of  $E_{corr}$ ,  $I_{corr}$  and  $R_{ct}$ , *indicating the similar good passivation effect of Al foil in PA solutions with different concentration in our experiments*. Since 30 wt% PA solution enabled the shortest Al foil-Ni55 layer separation time (5 mins), and thus was investigated as the typical example in our manuscript (Figure S8). It should be noted that Al-PA layer often exhibits much more dense structure in comparison to other metal-PA layers (e.g., Mg-PA layer), which could enable good passivation of Al foil under different PA solutions.R1

Reference:

R1 X. Cui, G. Jin, E. Liu, M. Ding, Q. Li and F. Wang, Mater. and Corros., 2012, **63**, 215.

The following statements have been added in the supplementary information (**See Page 4**):

*“To reveal the effect of PA concentration on the separation time and Al passivation, we conducted the Al foil-Ni55 layer separation in PA solutions with concentrations of 10, 20, 30, 40 and 50% (namely 10 wt% PA-Al foil, 20 wt% PA-Al foil, 30 wt% PA-Al foil, 40 wt% PA-Al foil and 50 wt% PA-Al foil), respectively. The as-separated S-Al foils were investigated by potentiodynamic polarization and impedance spectroscopy. As showed in Figure S6-7 and Table S3, all of separated Al (S-Al) foils demonstrated similar small values of  $E_{corr}$ ,  $I_{corr}$  and  $R_{ct}$ , indicating the similar good passivation effect of Al foil in the used PA solutions with different concentrations. Since 30 wt% PA solution enabled the shortest Al foil-Ni55 layer separation time (5 mins), and thus was investigated as the typical example (Figure S8).”*

Following figures have been added in the supplementary information (**Figure S6-7, Page 4**):

**Figure S6.** Tafel curves of (a) F-Al foil and S-Al foils separated by using (b) 10, (c) 20, (d) 30, (e) 40 and (f) 50 wt% PA solutions, respectively.

**Figure S7.** EIS plots of (a) F-Al foils and S-Al foils separated with (b) 10, (c) 20, (d) 30, (e) 40 and (f) 50 wt% PA solutions, respectively. The fitting was conducted based on the equivalent circuit model in Table S3.

The following table has been added in the supplementary information (**Table S3, Page 19**):

**Table S3.** Equivalent circuit and the corresponding values of  $R_{ct}$ .

| Types                         | Equivalent circuit                                                                 | $R_{ct}(\Omega)$ |
|-------------------------------|------------------------------------------------------------------------------------|------------------|
| 10 wt% PA-Al foil             |                                                                                    | 3782             |
| 20 wt% PA-Al foil             |                                                                                    | 3766             |
| 30 wt% PA-Al foil (S-Al foil) |  | 3793             |
| 40 wt% PA-Al foil             |                                                                                    | 3762             |
| 50 wt% PA-Al foil             |                                                                                    | 3798             |

3. Figures S3 and S4 do not illustrate separation time as claimed in the text on page 4!

**Response:** Thank you very much for careful comments. As suggested, separation time has been added in Figure S4 in the revised supplementary information. Figure S3 revealed the stoichiometric ratio of the degraded cathode active materials ( $\text{LiNi}_{0.55}\text{Co}_{0.15}\text{Mn}_{0.30}\text{O}_2$ ) without involving separation time.

The following statements have been revised in the manuscript (See Page 4, Line 101-102):

The initial statement “*Ni55 layer was completely separated from the Al foil in only 5 mins (Figure S3-5).*” has been replaced as “ *$\text{LiNi}_{0.55}\text{Co}_{0.15}\text{Mn}_{0.30}\text{O}_2$  (Ni55) layer was completely separated from the Al foil in only 5 mins (Figure S3-5 and Movie S1).*”

4. The authors make reference to the figures in the Supplementary information without previous explanation or description of the experiment undertaken. This leaves the reader to deduce the author’s intentions from the capture of the figures. In example-reference to Figure S9 presenting schematic diagrams and digital images of the spent Ni55 cathode with covered tap for PA penetration behavior investigation.

**Response:** We acknowledge the reviewer’s careful comments.

To reveal the reason for fast separation of Al foil and Ni55 layer, we sealed the top surface of Ni55 electrode, so that the PA solution could not penetrate from the top surface of the electrode, and only slight peeling of Al foil was shown on the edge of the electrode and the overall separation was hindered (Figure S11, Figure S9 in the initial supplementary information). These results indicated that the penetration of PA solution perpendicular to the electrode surface enabled the fast peeling of Ni55 layer from Al foil.

The following statements have been added in the supplementary information (See Figure S11, Page 6):

“*To reveal the reason for quick separation of Al foil and Ni55 layer, the top surface of a piece of Ni55 cathode was sealed, so that the PA solution could not penetrate from the top surface of the*

cathode, and only slight peeling of active material layer was shown on the edge of the cathode and the overall separation was hindered in sharp contrast to the fast peeling of the entire Ni55 layer from Al foil. These results indicated that the quick penetration of PA solution perpendicular to the electrode surface enabled the fast peeling of Ni55 layer from Al foil.”

5. The reference to Figure S12 on page 5 is entirely incorrect. Figure S12 illustrates optical microscopy images for Al foils separated by using ultrasonic treatment for 1 h in (a) DI water and (b) NMP solution. This is an experiment not discussed in the main text and certainly not the PA separation commented in the text.

**Response:** Thank you very much for pointing out this unclear statement. We conducted the separation of Al foil and Ni55 layer using widely used ultrasonic treatment as the control for the proposed separation approach in Figure S12. To avoid confusion, Figure S12 was removed in the revised manuscript.

6. Wrong figure reference on page 5:

“The Al residues in the separated Ni55 were as low as 0.026 wt%, again supporting the passivation of Al foil during the processing (Figure 2e and Table S2).” It should be referenced to Table S1!

**Response:** Sorry for this typo. We have corrected it in the revised manuscript (See Page 5, Line 127).

7. The lack of peaks indexation in Figures 2f and S16 makes the XRD analysis on page 5 unclear.

**Response:** We acknowledge the reviewer very much for this helpful suggestion. The peaks indexation has been added in the Figures 2f and S16. (See Figure 2f, Page 19 in manuscript. See Figure S16, Page 8 in supplementary information)

**Figures 2.** (f) XRD patterns of the separated Al foil and Ni55 layer.

**Figure S16.** XRD patterns of the Ni55 separated by using HCl solution.

8. The acronyms F-Al and S-Al are introduced in the text on pages 5 and 6 without explanation.

**Response:** Thank you very much for your careful comment. The acronyms F-Al and S-Al are denoted at their first appearance on Pages 4 and 6, as following:

“Also, the thickness of the separated Al foil (denoted as *S-Al foil*) was the same as that of the initial one (13  $\mu\text{m}$ ) ...” (See Page 4, Line 110)

“In comparison to fresh Al (*F-Al*) foil, new stretching vibration peaks at 1162, 1438 and 1641  $\text{cm}^{-1}$  were shown in the FT-IR spectrum of S-Al foil (Figure 3b) ...” (See Page 6, Line 148)

9. The term “degraded Ni55” is introduced without explanation on page 7 and its relation to Figure S19 is not clearly explained.

**Response:** Thank you very much for the careful comments. For clarity, the “*degraded Ni55*” was defined in the revised manuscript as “*Ni55 cathode from spent cell before Al foil-active material layer separation operation was denoted as degraded Ni55 for simplification.*” (See Page 8, Line 218-220). We also checked through the full manuscript and normalized its use.

As suggested, the relation between degraded Ni55 and Figure S28 (Figure S19 in the initial supplementary information) was explained in the revised manuscript as follows (See Page 8, Line 213-216):

“*Figure S28 showed the dissolved of Li, Ni, Co and Mn contents from the degraded Ni55 after its separation with Al foil in PA solution, and the results indicated that the dissolution of these elements was negligible (e.g., 0.27% for Ni, 0.08% for Co, and 0.17% for Mn).*”

10. The data on the electrochemical performance of regenerated and degraded Ni55 (Figures S24-28) should be commented in the text rather than simply referred to.

**Response:** We acknowledge the reviewer’s careful comments. For clarity, the mentioned figures have been discussed detailly in the supplementary information.

The following statements have been added in the supplementary information (See Figure S34-38, Page 14-16):

**For Figure S34** (Figure 24 in the initial supplementary information): “*The regenerated Ni55 showed lower interfacial resistance and charge transfer resistance than the degraded counterpart, which supported the successful materials recovery.*”

**For Figure S35** (Figure S25 in the initial supplementary information): “*As shown in the voltage profiles, the overpotential of the regenerated Ni55 was reduced under different charge rates than the degraded Ni55, which indicated the improvement of electrochemical reaction kinetics for the regenerated Ni55.*”

**For Figure S36** (Figure S26 in the initial supplementary information): “*After regeneration, the Ni55 displayed higher reversible capacities and lower potential hysteresis between charge and discharge curves at 0.3, 0.5, 1.0 and 2.0 C in comparison to the degraded Ni55, respectively.*”

**For Figure S37-38** (Figure S17-28 in the supplementary information): “*The CVs of the degraded and regenerated Ni55 demonstrated voltage differences of 0.33 and 0.22 V for the redox couples, respectively, which suggested enhanced electrochemical reaction kinetics of the Ni55 after regeneration.*”

11. The caption of Figure S25 does not explain clearly the presented graphs. I guess these are views of the charge curves zoomed to the lower specific capacity values. What was the goal of the authors presenting those?

**Response:** Thank you very much for your careful comments. As suggested, we have provided the detailed explanation for Figure S35 (Figure S25 in the initial supplementary information) in the revised manuscript (See the responses to Question 10). We presented this figure to show that the as-separated degraded Ni55 could be easily regenerated by facile annealing of its mixture with Li salt, supporting the advantage of maintaining structural integrity of Ni55 by the reaction-passivation mechanism driven Al foil-active material layer separation.

12. Simplified economic analysis of the recycling process (cost and availability) is desirable and can be done in comparison to a conventional separation process (e.g. vs HCl leaching).

**Response:** We would like to acknowledge for your valuable comments. As suggested, we conducted the simplified economic analysis for the PA involved direct recycling (PA-direct) process and the general hydrometallurgical (Hydro) process with EverBatt model developed by the Argonne National Laboratory (Figure R1-3). All components of the spent Ni55 cell should be crushed and annealed, followed by density separation, and degraded Ni55 was then leaching via reagents of HCl, H2O2 etc for Hydro process (Table R1). Their comparison of techno-economic analysis was conducted as following:

**Cost for processing.** The cost for labour and others is similar for PA-direct and Hydro approaches. However, Li salt is needed in the PA-direct for repairing cathode material (from the degraded Ni55 to regenerated Ni55), leading to relatively high cost of 6.78 \$ kg-1 (Figure R1a).

It is noted that more Li salt is needed for the re-synthesis of active cathode materials using the product in Hydro as raw materials, which is discussed later.

**Profit.** PA-direct possesses the advantage of high separation efficiency and production (>99.9 %) of cathode material and direct output of high-performance regenerated cathode material (Table R2). Thus, it brings higher revenue and profit of 15.79 and 9.01 \$ kg-1 in comparison to Hydro approaches (Figure R1b-c, 6.35 and 2.00 \$ kg-1 for revenue and profit of Hydro)

**Overall cost.** We performed the economic analysis based on manufacturing 1 kg-Ni55 cathode as a reference for industrial production process. It should be note due to the separating process of Hydro could not convert all the Ni55 materials to the corresponding salts, and thus additional Ni salt, Co salt, Mn salt and Li salt was needed for materials re-synthesis. Thus, the cost for Ni55 cathode produce via PA-direct was only 16.07 \$ kg-1, much lower than that via Hydro process (18.73 \$ kg-1 for Hydro, Figure R1d and Table R3).

As a result, our PA-direct provides a promising route for facile separation of metallic current collector and active material layer, and active materials regeneration, which is promising for practical applications.

To increase the impact of our work, the comprehensive LCA/TEA were conducted in the section of *Environmental and economic analysis of PA involved regeneration routine* in Page 9-10 (See the manuscript).

**Figure R1.** (a) Cost, (b) revenue, (c) profit and (d) overall cost of manufacturing 1 kg-Ni55 cathode by PA-direct and Hydro.

**Figure R2.** Process diagram of PA-direct.

**Figure R3.** Process diagram of Hydro.

**Table R1.** Materials requirements for 1 kg-spent cells recycling through PA-direct and Hydro approaches.

| Hydro             | PA-direct         |
|-------------------|-------------------|
| Hydrochloric Acid | Lithium Hydroxide |
| Hydrogen Peroxide | PA solution       |
| Sulfuric Acid     |                   |
| Soda Ash          |                   |
| Sodium Hydroxide  |                   |

**Table R2.** Prices of materials for PA-direct and Hydro approaches.

|                                 | Price (\$ t -1 ) | Data sources |
|---------------------------------|-----------------------------|--------------|
| PA solution (50 wt%)            | 5764.20                     | Table S10    |
| Ni55                            | 32683.54                    | 10jqka       |
| LiOH·H 2 O           | 42561.23                    | SMM          |
| Li 2 CO 3 | 32756.17                    | SMM          |
| graphite                        | 4459.49                     | SMM          |
| Al foil                         | 2396.79                     | SMM          |
| Cu foil                         | 10110.11                    | SMM          |
| HCl                             | 8.71                        | 100ppi       |
| Ammonia                         | 459.99                      | 100ppi       |
| Ammonium Bicarbonate            | 196.10                      | 100ppi       |
| Hydrogen Peroxide               | 115.24                      | 100ppi       |
| Ni in product                   | 23355.19                    | SMM          |
| Mn in product                   | 2742.69                     | SMM          |
| Co in product                   | 30666.03                    | SMM          |
| Water                           | 1.76                        | BDB          |
| Sewage treatment                | 2.20                        | 51wctt       |

**Data sources:**

SMM (<https://www.smm.cn/>), 10jqka (<https://www.10jqka.com.cn/>), 100ppi (<https://www.101ppi.com/>) and BDB (<http://sz.bendibao.com/>). The data was collected in May 2023.

**Table R3.** The TEA of manufacturing 1 kg-cell from raw and degraded Ni55 materials.

|                  | Cost (\$ kg -1 ) |              |            | Price of Ni55 (\$ kg -1 ) | Profit (\$ kg -1 ) |
|------------------|-----------------------------|--------------|------------|--------------------------------------|-------------------------------|
|                  | Recycling                   | Regeneration | Production |                                      |                               |
| Hydro     | 10.87                       | NA           | 7.86       | 32.68                                | 13.95                         |
| PA-direct | NA                          | 16.07        | NA         | 32.68                                | 16.80                         |

13. It is unclear from the text in the manuscript how referencing to sources [32-34] contribute to the claims made by the authors?

**Response:** Thank you very much for your valuable comment. We carefully checked the as-mentioned references and revised accordingly.

Reference [32,34] was deleted. References [33] illustrated that the regenerated Ni-rich cathode showed improved potential hysteresis in comparison to the counterpart before materials recovery, which was consisted with our results, thus it was cited after the statement “*The successful regeneration of the degraded Ni55 was also evidenced by the results of EIS, constant-current charge/discharge test and cyclic voltammetry (Figures S34-38 and Table S3-4)36*”.

## **REVIEWERS' COMMENTS**

### **Reviewer #1 (Remarks to the Author):**

The authors have carefully revised the manuscript to address the technical comments by the authors.

The main issue remains on its significance for making an impact on the critical pathways of lithium-ion batteries (e.g. the Li, Ni, Co, or even Mn) - as Al is not as critical as the other elements.

While the technical quality and the scientific findings are of interest to the topical readership of recycling, it is still unclear if the impact and overall novelty are commensurate with Nature Communications.

### **Reviewer #2 (Remarks to the Author):**

This is an extremely comprehensive rewrite of the manuscript in which the authors have gone beyond the comments raised by the reviewers to improve their manuscript. I think that the paper is suitable for publication and this is a very good example of where good refereeing and authors who take the comments seriously can deliver a significantly improved paper. Well done.

### **Reviewer #3 (Remarks to the Author):**

The authors answered all questions related to previously marked mistakes and inconsistencies. The deficiencies in the text were addressed by numerous additional contributions which made the presentation of the authors results and their concept more clear and convincing. The demonstration of strong covalent bonding via numerous characterization techniques added weight to the main claim in the manuscript, namely reaction-passivation mechanism of the cathode layer de-bonding. LCA and TEA analyses made the manuscript relevant to a wider reader audience.

Thus, I recommend as-revised manuscript to be accepted for publication.

## **Response to reviewer' comments:**

### ***Reviewer #1***

Comments:

The authors have carefully revised the manuscript to address the technical comments by the authors. The main issue remains on its significance for making an impact on the critical pathways of lithium-ion batteries (e.g. the Li, Ni, Co, or even Mn) - as Al is not as critical as the other elements. While the technical quality and the scientific findings are of interest to the topical readership of recycling, it is still unclear if the impact and overall novelty are commensurate with Nature Communications.

**Response:** We thank the reviewer very much for satisfying with our revision about all the technical concerns. Regarding to the concern about the significance and impact of our work based on the statements “*Al is not as critical as the other element...*”, we emphasize that main achievement of our work does not rely on the recycling of Al foil, and conclude the impact and novelty of our work as follows: “Facile and efficient separation of active material layer from Al foil becomes one of the most important procedures among all the operations for battery cathode recycling. This work explored a novel reaction-passivation mechanism driven separation of Al foil and active material layer of cathodes, and first demonstrated complete separation of Al foil and cathode active material layer with ultrahigh separation efficiency (>99.9%) for an entire spent 102 Ah-level  $\text{LiNi}_{0.55}\text{Co}_{0.15}\text{Mn}_{0.3}\text{O}_2$  (Ni55)||graphite cell from electric vehicle within 5 mins. The separated Ni55 with low Al impurity and negligible damage in material structure provided the great bonus for facile regeneration of degraded Ni55 via direct annealing, and the regenerated Ni55 delivered comparable performance to the stat-of-the-art fresh Ni55 cathode material.” Please also see more details in the general response part to Reviewer #1 (the last responses) and the contents in the main text (Line 206-226, Page 8; Figure 4, Page 22).

### ***Reviewer #2***

This is an extremely comprehensive rewrite of the manuscript in which the authors have gone beyond the comments raised by the reviewers to improve their manuscript. I think that the paper is suitable for publication and this is a very good example of where good refereeing and authors who take the comments seriously can deliver a significantly improved paper. Well done.

**Response:** We thank the reviewer very much for his/her very positive evaluation.

### ***Reviewer #3***

The authors answered all questions related to previously marked mistakes and inconsistencies. The deficiencies in the text were addressed by numerous additional contributions which made the presentation of the authors results and their concept more clear and convincing. The demonstration of strong covalent bonding via numerous characterization techniques added

weight to the main claim in the manuscript, namely reaction-passivation mechanism of the cathode layer de-bonding. LCA and TEA analyses made the manuscript relevant to a wider reader audience. Thus, I recommend as-revised manuscript to be accepted for publication.

**Response:** We appreciate the positive comments very much by the reviewer and his/her support for publication of our work in Nature Communications.